# Shared requirement for MYC upstream super-enhancer region in tissue regeneration and cancer

Inderpreet Sur[1], Wenshuo Zhao[1], Jilin Zhang[1], Margareta Kling Pilström[1], Anna T Webb[2], Huaitao Cheng[3], Ari Ristimäki[5,6], Pekka Katajisto[2,7,8], Martin Enge[3], Helena Rannikmae[4], Marc de la Roche[4], Jussi Taipale[1,5,9]

Cancer has been characterized as a wound that does not heal. Malignant cells are morphologically distinct from normal proliferating cells but have extensive similarities to tissues undergoing wound healing and/or regeneration. The mechanistic basis of this similarity has, however, remained enigmatic. Here, we show that the genomic region upstream of *Myc*, which carries more cancer susceptibility in humans than any other genomic region, is required for intestinal regeneration after radiation damage. Failure to regenerate is associated with inefficient *Ly6a*/Sca1+ stem/progenitor cell mobilization, and almost complete failure to re-establish Lgr5+ cell compartment in the intestinal crypts. The *Myc* upstream region is also critical for growth of adult intestinal cells in 3D organoid culture. We show that culture conditions recapitulating most aspects of adult normal tissue architecture still reprogram normal cells to proliferate using a mechanism similar to that employed by cancer cells. Our results establish a function for the $Myc^{2-540}$ super-enhancer region as the genetic link between tissue regeneration and tumorigenesis, and demonstrates that normal tissue renewal and regeneration of tissues after severe damage are mechanistically distinct.

## Introduction

Out of all genomic regions, the ~500-kb super-enhancer region on 8q24 that regulates MYC expression carries the largest burden of population-level cancer susceptibility (Sur et al, 2013; Buniello et al, 2019); it contains several common alleles that increase the risk for multiple major forms of human cancer (Gudmundsson et al, 2007, 2009, 2012; Yeager et al, 2009; Buniello et al, 2019). For example, the cancer-associated G allele of SNP rs6983267 is found in 50% and >85%

of people with European and African ancestry, respectively (Haiman et al, 2007; Yeager et al, 2007). This allele alone increases the risk for colorectal and prostate cancers more than 20%, causing hundreds of thousands of cancer deaths globally per year (https://www.wcrf.org/cancer-trends/global-cancer-data-by-country/). Although this highly conserved super-enhancer region is known to regulate MYC expression in specific tumors, its normal function has not been established. We have previously deleted regulatory sequences syntenic to the super-enhancer region in mice (Dave et al, 2017) and demonstrated that the mice are healthy and fertile despite greatly reduced levels of MYC expression in epithelia of the breast, prostate, and intestine. The mice, however, are highly resistant to intestinal and mammary tumors. Although deletion of the region did not cause a major effect on fitness in our assays, the enhancer elements within it are highly conserved between mammalian species (Hallikas et al, 2006; Pomerantz et al, 2009; Tuupanen et al, 2009; Ahmadiyeh et al, 2010; Sotelo et al, 2010; Hnisz et al, 2013; Yan et al, 2013). This suggests that this region would also have a beneficial biological function that would balance the negative impact of the increased cancer predisposition.

Because MYC is one of the main drivers of cell proliferation, we hypothesized that its upstream super-enhancer region would be beneficial under some environmental condition that requires rapid cell proliferation, but which the mice fed and housed in a clean laboratory environment would not encounter. For example, the super-enhancer region could be important for responses to chronic infections, wounding, or other forms of tissue damage (Schafer & Werner, 2008; Plummer et al, 2016). In this work, we demonstrate that the regulatory region $Myc^{2-540}$ functions as the genetic link between tissue regeneration and tumorigenesis and establish that mechanisms of homeostatic cell proliferation that maintain the stable number of stem cells and their progeny are distinct from mechanisms used during the growth of intestinal organoids, healing tissues, and tumors, all of which require an exponential increase in the number of stem cells.

[1]Department of Medical Biochemistry and Biophysics, Karolinska Institutet, Stockholm, Sweden   [2]Department of Cell and Molecular Biology, Karolinska Institutet, Stockholm, Sweden   [3]Department of Oncology and Pathology, Karolinska Institutet, Stockholm, Sweden   [4]Department of Biochemistry, University of Cambridge, Cambridge, UK   [5]Applied Tumor Genomics Program, Biomedicum, University of Helsinki, Helsinki, Finland   [6]Department of Pathology, HUSLAB, HUS Diagnostic Center, Helsinki University Hospital and University of Helsinki, Helsinki, Finland   [7]Institute of Biotechnology, HiLIFE, University of Helsinki, Helsinki, Finland   [8]Molecular and Integrative Bioscience Research Programme, Faculty of Biological and Environmental Sciences, University of Helsinki, Helsinki, Finland   [9]Generative and Synthetic Genomics Programme, Wellcome Sanger Institute, Hinxton, UK

Correspondence: jussi.taipale@ki.se
Helena Rannikmae's present address is Complex In Vitro Models, In vitro/In Vivo Translation, GlaxoSmithKline, UK

 

# Results

## $Myc^{\Delta 2\text{-}540/\Delta 2\text{-}540}$ mice fail to recover from radiation-induced intestinal damage

In order to investigate whether the super-enhancer region was required for tissue repair and regeneration, we subjected the $Myc^{\Delta 2\text{-}540/\Delta 2\text{-}540}$ mice to ionizing radiation, which causes uniform and highly reproducible damage to the intestinal lining (Kim et al, 2017; Gu et al, 2020). WT mice γ-irradiated for a total dose of 12 Gy rapidly developed intestinal damage and lost 7.0 ± 0.8% of their weight in 3 d (n = 19); the weight loss stabilized at that time, and the damage was largely repaired by day 5. In contrast, the $Myc^{\Delta 2\text{-}540/\Delta 2\text{-}540}$ mice appeared unable to repair the damage, as they continued to lose weight beyond 3 d, losing on average 17% by 5 d (n = 10) (Fig 1A and B) at which time they had to be euthanized for humane reasons. Morphological analysis of the small intestines confirmed that the damage was repaired in the WT mice but sustained in the $Myc^{\Delta 2\text{-}540/\Delta 2\text{-}540}$ mice because of failure to regenerate the intestinal crypts (Figs 1C and D and S1A). The WT mice displayed a strong proliferative response at both 3 and 5 d after irradiation, whereas at the same time points, in the $Myc^{\Delta 2\text{-}540/\Delta 2\text{-}540}$ mice there was almost complete absence of proliferating cells (Fig 1E and F). The proliferative response was accompanied by the induction of MYC mRNA expression. Although irradiation induced MYC expression in the WT, with peak induction of $Myc$ transcripts occurring at 3 days post-irradiation (dpi, Figs 2A and B and S1B and D), the number of Ki-67⁺ cells remained high at 5 dpi even when the MYC transcripts returned to pre-radiation levels. In $Myc^{\Delta 2\text{-}540/\Delta 2\text{-}540}$ mice, $Myc$ levels were almost undetectable both before and after irradiation (Figs 2A and B and S2D). The cellular architecture and expression profile of the small intestine of the $Myc^{\Delta 2\text{-}540/\Delta 2\text{-}540}$ mice and WT were similar before irradiation (Fig S2A–C), suggesting that loss of proliferation was specific to the repair/regenerative stimulus. To decipher changes in cellular transcriptome when MYC levels increase after tissue damage, we performed single-cell RNA-seq (scRNA-seq) analysis at day 2 after irradiation. Analysis of scRNA-seq data revealed that $Myc$ expression was limited to stem and transit amplifying cells of WT mice. Despite the irradiation, which robustly induced MYC in WT mice, stem cells or transit amplifying cells from $Myc^{\Delta 2\text{-}540/\Delta 2\text{-}540}$ mice expressed almost no $Myc$, the signal being even lower than that from irradiated mature enterocytes of WT mice (Fig 2C). The $Myc^{\Delta 2\text{-}540/\Delta 2\text{-}540}$ mice were, however, fully capable of mounting a normal DNA damage response as seen by the induction of p53 target genes $Cdkn1a$ (p21) and $Bbc3$ (PUMA), which facilitate p53-dependent cell cycle arrest and apoptosis, respectively (Figs 2D and S1C). These results show that despite a normal tissue architecture, $Myc^{\Delta 2\text{-}540/\Delta 2\text{-}540}$ mice fail to repair tissue after damage induced by ionizing radiation.

## Failure to regenerate is associated with loss of induction of MYC

To determine whether the loss of regenerative potential seen in $Myc^{\Delta 2\text{-}540/\Delta 2\text{-}540}$ intestines was due to MYC, we analyzed the expression of all genes within 5 Mb of $Myc$. By far, the strongest effect seen was for $Myc$ itself, whose expression was almost completely abolished in the mutant intestine. Transcripts of some genes telomeric of $Myc$ were also down-regulated, but to a much lesser extent. These include the noncoding RNA $Pvt1$ and the gasdermin C genes $Gsdmc2$, $Gsdmc3$, and $Gsdmc4$ (Table S1). However, only the expression of $Myc$ and $Pvt1$ was specifically up-regulated in WT cells upon irradiation. Because $Pvt1$ lncRNA is thought to respond to the same enhancer elements and elicit its effect via interaction with $Myc$ (Tseng et al, 2014; Cui et al, 2016), the results indicate that MYC is the primary effector of the $Myc^{\Delta 2\text{-}540/\Delta 2\text{-}540}$ phenotype. Consistent with this, unbiased gene set enrichment analysis revealed that a set of MYC targets (V1) failed to be up-regulated by irradiation in stem cells of the $Myc^{\Delta 2\text{-}540/\Delta 2\text{-}540}$ mice (Fig 3A, Table S2). We also specifically analyzed the expression of a set of 126 functionally conserved MYC targets from Zielke et al (2022), most of which are involved in metabolism and ribosome biogenesis. The regulation of this set of genes is conserved between $Drosophila$ and humans with most (~90%) of the genes being MYC-inducible. These functionally conserved MYC targets were also strongly up-regulated in the stem cells of irradiated WT mice compared to $Myc^{\Delta 2\text{-}540/\Delta 2\text{-}540}$ mice (Fig 3B, Table S3) with 64% of the genes showing a significant ($P < 0.05$) decrease in $Myc^{\Delta 2\text{-}540/\Delta 2\text{-}540}$ cells. We also checked the expression of known MYC-repressed target genes (Diamant et al, 2025); however, these did not show a marked change in expression after irradiation (Fig 3C, Table S4). Taken together, the results establish that induction of MYC drives a robust program of increased ribosome biogenesis that can facilitate growth of new tissue to repair the damage caused by the irradiation. However, among the MYC-regulated genes, we also observed induction of the negative translational regulator eukaryotic initiation factor 4E-binding protein 1 ($Eif4ebp1$/4EBP1); Tameire et al, 2019; Fig 3D), suggesting that induction of MYC increases translational capacity, through its effect on ribosomal biogenesis, but is not sufficient to induce translational activity. An increase in translational activity will require additional signaling, such as activation of mTOR complex (mTORC) kinase that can phosphorylate and inactivate 4EBP1 (Saxton & Sabatini, 2017). Consistent with this model, it has been previously shown that mTORC signaling is required for regeneration of the intestine after radiation damage (Sampson et al, 2016). We also examined the expression of mTOR in our scRNA-seq data after irradiation in $Myc^{\Delta 2\text{-}540/\Delta 2\text{-}540}$ and WT cells. At 2 dpi, the WT cells express only slightly higher levels of mTOR mRNA compared with $Myc^{\Delta 2\text{-}540/\Delta 2\text{-}540}$ cells ($\log_2$ FC = 0.246 in stem cells) (Table S5). Such a small change in mRNA is unlikely to explain changes in levels of $Eif4ebp1$ between these conditions. Instead, the change is consistent with MYC induction of $Eif4ebp1$ mRNA combined with TOR-dependent phosphorylation of the 4EBP1 protein (Hruby et al, 2024 Preprint; Tameire et al, 2019). IHC staining of the intestines of both WT and $Myc^{\Delta 2\text{-}540/\Delta 2\text{-}540}$ mice before radiation damage revealed a strong signal for 4EBP1 and phosphorylated 4EBP1 (Thr37/46), whereas after radiation (3 dpi), staining was more prominent in the regenerating crypts of the WT intestine (Fig 3E and F) consistent with a role of both MYC and mTORC in the regenerative response.

We next asked whether developmental or homeostatic changes that may have occurred before radiation contributed to the failure of regeneration; we also performed scRNA-seq on the crypt-enriched epithelium of nonirradiated (non-IR) WT and $Myc^{\Delta 2\text{-}540/\Delta 2\text{-}540}$ mice. Analysis of cell types by scRNA-seq and immunohistochemistry of

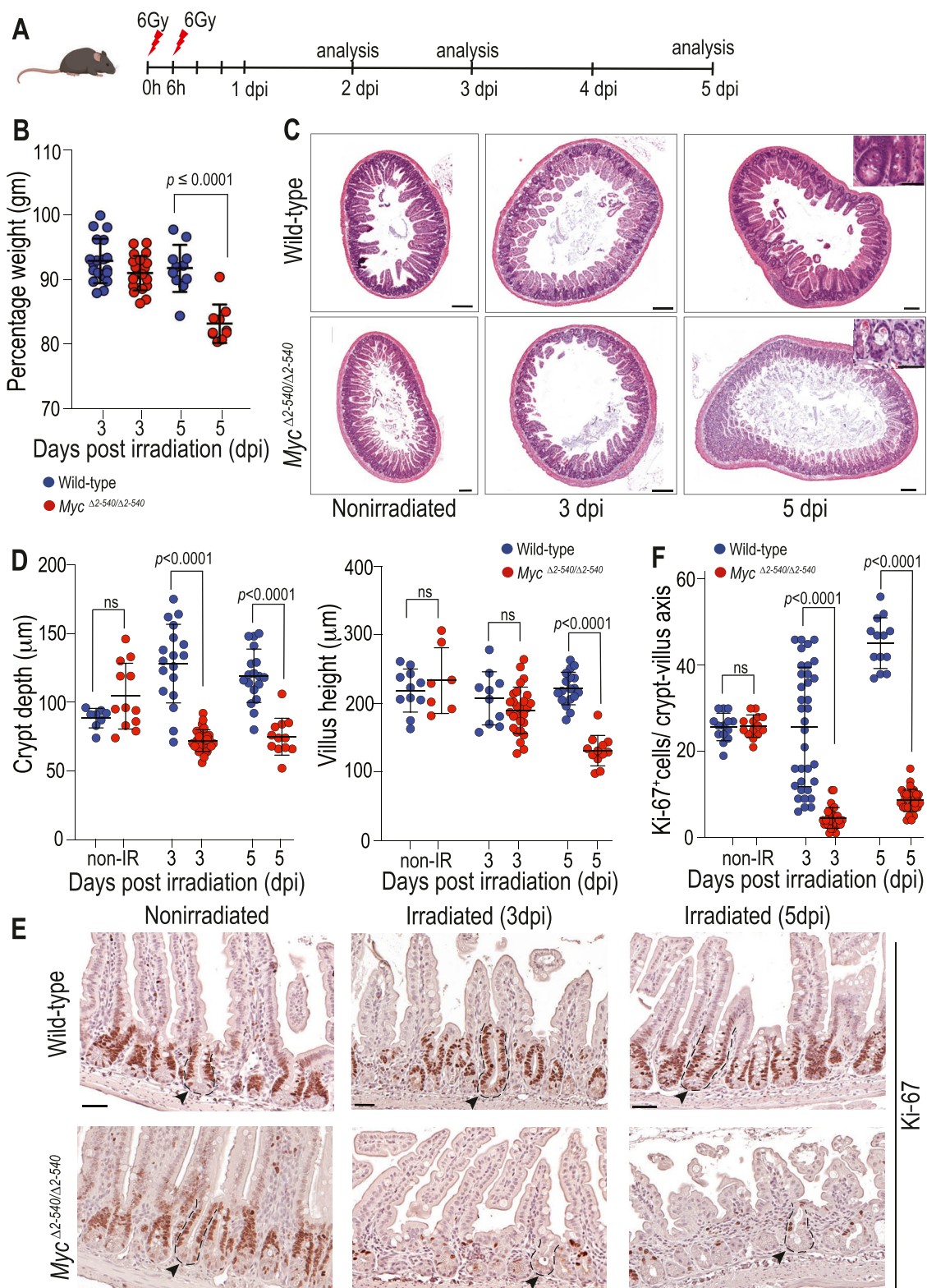

**Figure 1.** *Myc^{Δ2-540/Δ2-540}* **intestine fails to regenerate in vivo post–γ-irradiation.**
**(A)** Timeline of the irradiation experiment. **(B)** *Myc^{Δ2-540/Δ2-540}* mice (red) are more sensitive to weight loss post-irradiation compared with WT mice (blue). Percentage weight relative to the weight before irradiation is shown. WT: n = 19 (3 dpi), n = 11 (5 dpi), *Myc^{Δ2-540/Δ2-540}*: n = 21 (3 dpi), n = 10 (5 dpi). **(C)** Hematoxylin-and-eosin (H&E)–stained sections of small intestine showing *Myc^{Δ2-540/Δ2-540}* mice fail to regenerate crypts after radiation-induced damage (also see Fig S1A). **(D)** Crypt depth and villus height were significantly decreased in *Myc^{Δ2-540/Δ2-540}* mice compared with the WT after radiation. Longitudinally cut crypt and villi were measured from at least

markers of cell differentiation and proliferation before damage revealed no major differences between WT and $Myc^{\Delta2-540/\Delta2-540}$ mice (Fig S2A–C). Despite the lack of a major difference in proliferation and histological markers, there were cell type–specific differences in gene expression between WT and $Myc^{\Delta2-540/\Delta2-540}$ mice (Table S6). The top genes in the differentially expressed gene list ranked based on average expression (Fig S3) were genes involved in ribosome biogenesis (Table S6). Furthermore, in the normal nonirradiated $Myc^{\Delta2-540/\Delta2-540}$ mice, the functionally conserved MYC targets were also modestly down-regulated within the stem and transit amplifying cells compared with WT mice (Fig 3B, Table S3). In contrast, there was no down-regulation of the MYC target genes in mature enterocytes, which do not express MYC. Single-cell ATAC-seq analysis confirmed the absence of compensatory changes in accessibility of $Myc$ enhancer elements around the 538-kb enhancer region deleted in $Myc^{\Delta2-540/\Delta2-540}$ mice (Fig S2E). However, we observed an increase in $Mycl$ expression within the stem and transit amplifying cell clusters of $Myc^{\Delta2-540/\Delta2-540}$ mice suggesting compensation of MYC function during homeostasis (Fig S2F–H). It is noteworthy that the functionally conserved MYC target gene set that we find enriched after tissue damage was originally detected from tumor cells (Zielke et al, 2022), suggesting that increased ribosomal biogenesis and metabolic flux drive both cancer and tissue repair. These results indicate that the regenerative failure of the $Myc^{\Delta2-540/\Delta2-540}$ is most likely due to failure to respond to damage, as opposed to differences in tissue structure, morphology, or gene activity before the damage.

### Regeneration defects of adult and embryonic $Myc^{\Delta2-540/\Delta2-540}$ intestines in 3D cultures

To further characterize the role of the $Myc^{2-540}$ region in tissue repair, we decided to test a recently described ex vivo tissue regeneration assay, where mouse ileum is decellularized with sodium deoxycholate resulting in a matrix reminiscent of the 3D intestinal structure in vivo that contains the distinct extracellular matrix structures where former crypts and villi were located. Overlaying this decellularized extracellular matrix (dECM) with isolated crypts results in repopulation of the empty crypt pits and regeneration of the entire intestinal epithelium (Iqbal et al, 2025). In this assay, cultures of WT crypts repopulated the dECMs and gave rise to crypt and villus structures, whereas crypts from $Myc^{\Delta2-540/\Delta2-540}$ mice failed to do so (Fig 4A). As repopulation of the bare dECM requires cellular motility, with proliferation and self-renewal restricted to the empty crypt pits, we next placed isolated crypts in standard organoid culture conditions in Matrigel. Surprisingly, adult intestinal organoids from $Myc^{\Delta2-540/\Delta2-540}$ mice failed to grow also in such culture, where self-renewal and proliferation-promoting signals are provided evenly (Fig 4B).

Establishment of organoid culture from adults, like tissue repair, is associated with a change towards a more fetal-like state (Nusse

et al, 2018; Yui et al, 2018). We therefore tested whether we could establish 3D cultures from intestinal epithelium/crypts of E16.5 $Myc^{\Delta2-540/\Delta2-540}$ embryos. In contrast to adults, 3D cultures from $Myc^{\Delta2-540/\Delta2-540}$ embryos did grow, forming spheroids. However, unlike WT cultures, the $Myc^{\Delta2-540/\Delta2-540}$ spheroids failed to generate budding mature crypt-like structures during the first 10 d of culture (Fig 4C). After extended passaging of the cultures for more than 2 wk, the spheroid growth pattern was often lost, and budding organoids, with mature crypt-like structures, formed also in the $Myc^{\Delta2-540/\Delta2-540}$ 3D cultures (Fig 4D). Henceforth, the organoids, with mature crypt-like structures, are referred to as budding organoids. We confirmed that $Myc$ was dramatically reduced in $Myc^{\Delta2-540/\Delta2-540}$ spheroids and late-budding organoids, both of which expressed less than 5% of the WT $Myc$ levels (Fig 4E). These results suggest that mechanism/s independent of MYC can drive embryonic intestinal growth in culture. Because previous studies have shown that both MYC and N-MYC are expressed in the developing mouse intestine at this stage, we also measured $Mycn$ expression from the organoid cultures, and found that $Myc^{\Delta2-540/\Delta2-540}$ organoid cultures expressed ~3.5-fold higher levels of $Mycn$ compared with the WT (Fig 4E), potentially compensating for the loss of MYC. To confirm that the failure of the $Myc^{\Delta2-540/\Delta2-540}$ embryonic organoids to bud was due to MYC loss and not other effects caused by the loss of the regulatory element, we treated WT embryonic organoids with the MYC inhibitor 10058-F4. This compound has previously been shown to induce a MYC loss–dependent diapause in early mouse embryos (Scognamiglio et al, 2016). Treatment of WT embryonic organoids during passage 1, when they are already budding, with 100 $\mu$M of 10058-F4 resulted in a clear shift from a budding to a spheroidal growth pattern (Fig S4). Despite the difference in culture, the WT and $Myc^{\Delta2-540/\Delta2-540}$ intestines of E16.5 embryos were morphologically comparable in vivo (Fig 4F), which is in line with our observation from the adult intestine that WT and $Myc^{\Delta2-540/\Delta2-540}$ are histologically and morphologically comparable during homeostasis. Taken together, these results establish that although the $Myc$ upstream super-enhancer region is not required for normal tissue renewal in mouse intestine, it is required for normal in vitro maturation of embryonic organoid cultures, and for growth of adult intestinal cells under organoid culture conditions.

### $Myc^{\Delta2-540/\Delta2-540}$ mice fail to mobilize $Ly6a$ (Sca1)$^+$ stem/progenitor cells during regeneration

To identify the mechanism underpinning the failure of $Myc^{\Delta2-540/\Delta2-540}$ intestines to regenerate, we determined whether the loss of this region affected the mobilization of reserve stem/progenitor cells, which previously have been shown to be associated with intestinal regeneration (Nusse et al, 2018; Yui et al, 2018). We performed scRNA-seq on cells from 11-d-old organoid cultures derived from E16.5 $Myc^{\Delta2-540/\Delta2-540}$ and WT embryos. At this time point, the 3D

---

two mice per category. Each dot represents an individual crypt/villus. **(E, F)** Failure to regenerate the crypts is associated with loss of proliferation as shown by IHC for the proliferation marker Ki-67. Ki-67$^+$ cells were counted within the crypt–villus axis from at least two mice per category (WT: nonirradiated [non-IR], n = 2; 3 dpi, n = 4; 5 dpi, n = 4 mice; $Myc^{\Delta2-540/\Delta2-540}$: non-IR, n = 2; 3 dpi, n = 3; 5 dpi, n = 3). Each dot represents the number of Ki-67$^+$ cells in an individual crypt–villus. Statistical significance was calculated using an unpaired $t$ test (two-tailed), and corresponding $P$-values are shown. A nonsignificant statistical difference is marked by ns. Error bars denote the mean ± SD. Magnification bars in C = 200 $\mu$m (inset = 50 $\mu$m), E = 50 $\mu$m. Representative crypts in E are outlined with dashed lines and marked with arrowheads.

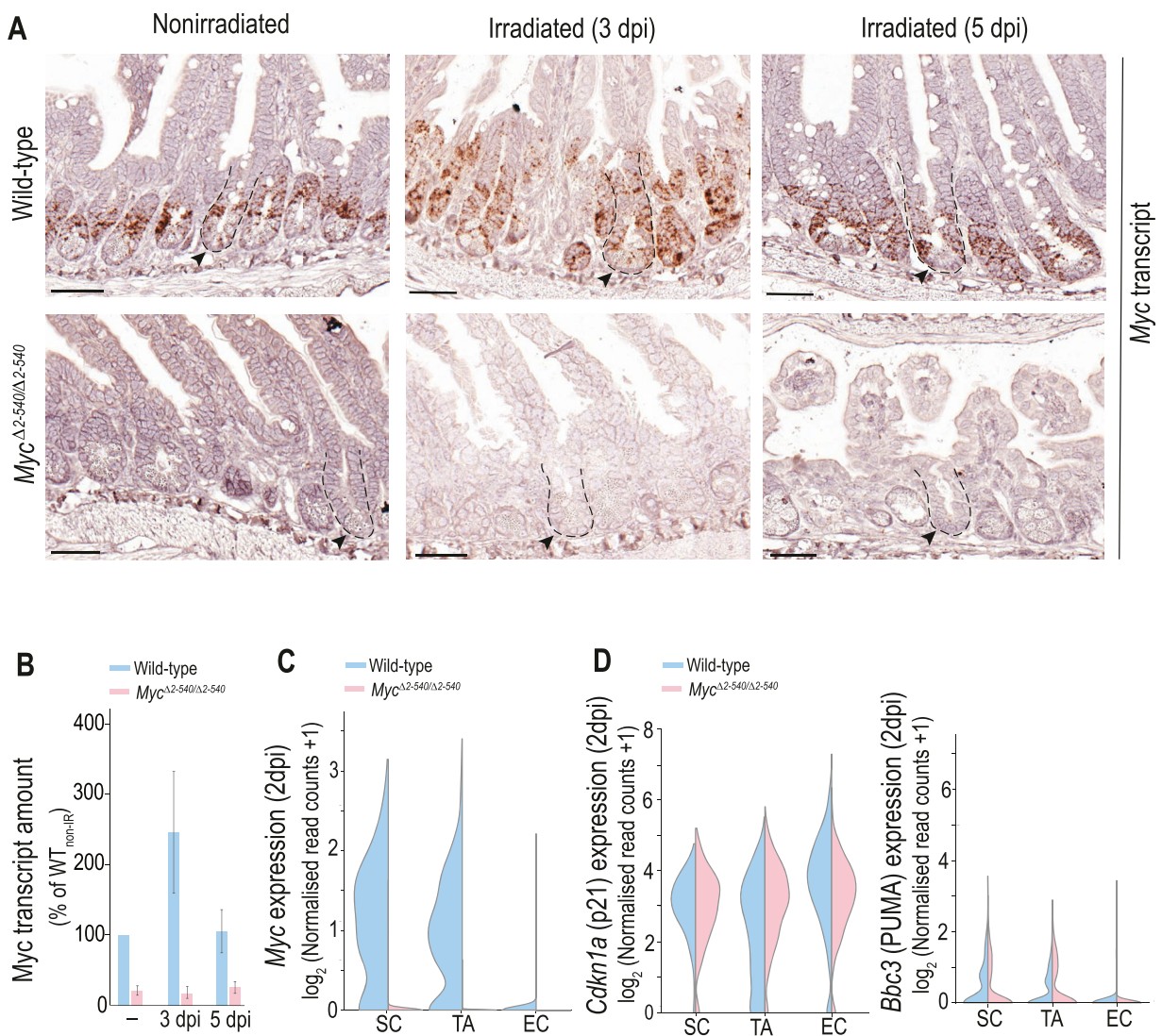

**Figure 2.** $Myc^{\Delta2-540/\Delta2-540}$ intestine is unable to up-regulate $Myc$ expression after irradiation.
**(A, B, C)** $Myc$ expression (A) ISH; (B) qPCR, WT: n = 2 (non-IR), n = 3 (3 dpi), n = 3 (5 dpi); $Myc^{\Delta2-540/\Delta2-540}$: n = 2 (non-IR), n = 2 (3 dpi), n = 3 (5 dpi); and (C) scRNA-seq (violin plots depicting the expression of $Myc$ in SC, TA, and EC populations) show almost complete absence of $Myc$ transcripts after radiation. **(D)** scRNA-seq, violin plots showing that unlike $Myc$ levels, the levels of p53-mediated DNA damage response indicators $Cdkn1a$ (p21) and $Bbc3$ (PUMA) were comparable between WT and $Myc^{\Delta2-540/\Delta2-540}$ intestine at 2 dpi (scRNA-seq). See also Fig S1. SC: stem cell; TA: transit amplifying cell; EC: enterocyte. Magnification bars in A = 50 $\mu$m. Representative crypts in A are outlined with dashed lines and marked with arrowheads.

cultures from WT contain organoids with distinct budding morphology, whereas cultures from $Myc^{\Delta2-540/\Delta2-540}$ consist of large spheroids and are devoid of mature crypt-like buds. scRNA-seq analysis showed that although spheroids from $Myc^{\Delta2-540/\Delta2-540}$ contained adult differentiated cell types including enterocytes and goblet cells (Fig S5), the stem cell compartment expressed low levels of canonical stem cell marker genes including $Lgr5$ (Fig S5). We performed additional clustering of the WT and $Myc^{\Delta2-540/\Delta2-540}$ stem cell population to determine its substructure. This identified five stem cell subclusters (SSC1F–SSC5F) (Fig 5A and B) in which the expression of proliferation markers was only minimally affected by $Myc^{2-540}$ deletion (Tables S7 and S8). However, cells of clusters SSC1F and SSC3F were enriched in WT cultures and expressed adult stem cell markers including $Lgr5$. $Myc^{\Delta2-540/\Delta2-540}$ cultures were instead

enriched in cells belonging to subclusters SSC2F, SSC4F, and SSC5F. These subclusters mainly expressed cell markers associated with fetal and regenerative states (Nusse et al, 2018; Yui et al, 2018) (Fig 5C, Tables S7 and S8).

Despite expressing several fetal markers, $Myc^{\Delta2-540/\Delta2-540}$ stem cells had much lower expression of the fetal/regenerative marker $Ly6a$ (Sca1) compared with WT (Fig 5C). Because the expression of $Ly6a$ (Sca1) is known to be induced by colitis and radiation-induced damage, as well as damage caused by helminth infections (Nusse et al, 2018; Yui et al, 2018), we checked whether the expression of $Ly6a$ (Sca1) was also affected in vivo in $Myc^{\Delta2-540/\Delta2-540}$ mice after radiation damage. Analysis of our scRNA-seq data showed that irradiation of the WT intestine mobilized a robust population of $Ly6a$ (Sca1)-expressing cells. These cells were distributed among

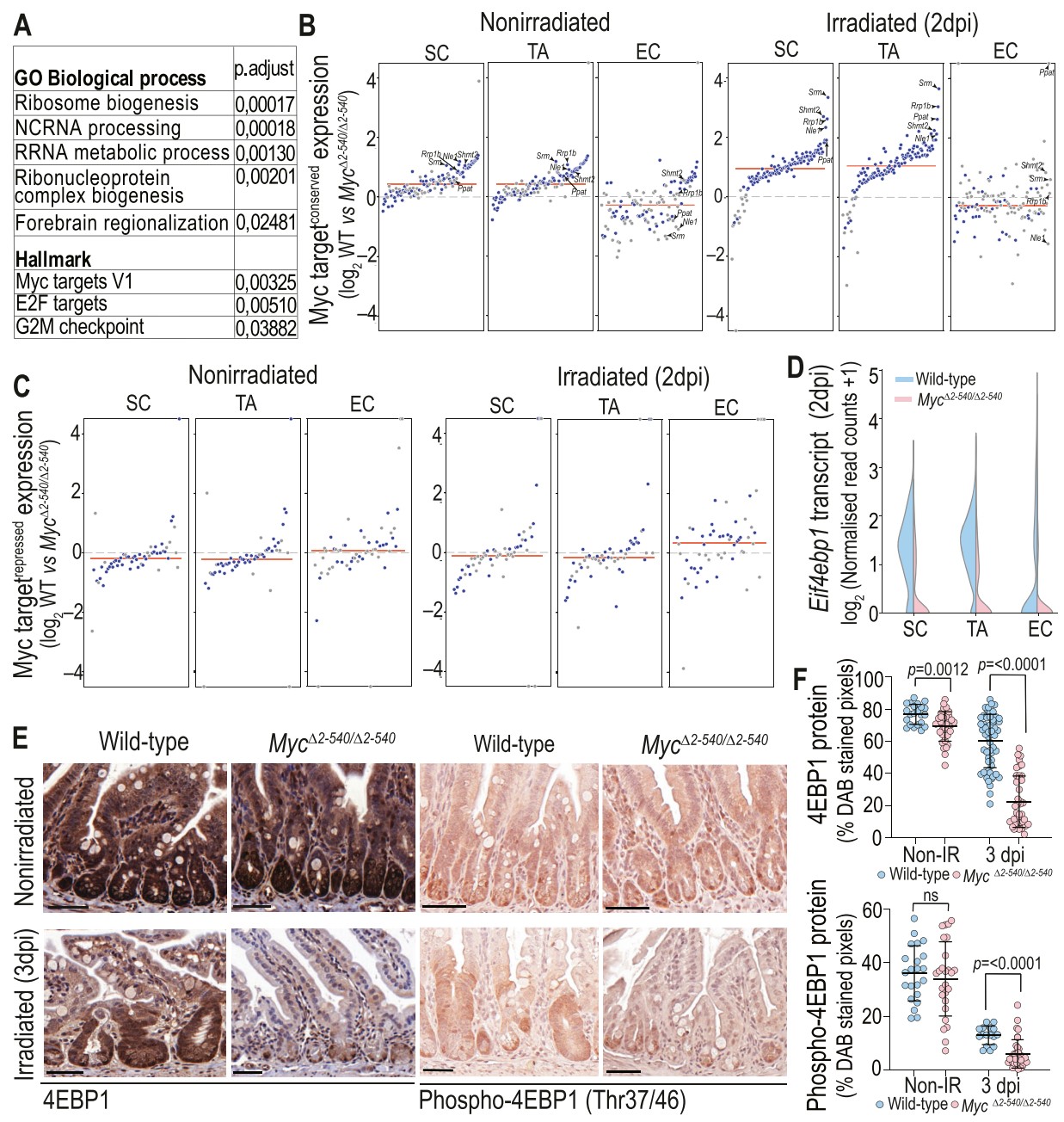

**Figure 3.** *Myc^{Δ2-540/Δ2-540}* **intestine fails to induce MYC targets after irradiation, compromising mTORC signaling.**
**(A)** GSEA showing the most enriched Gene Ontology and Hallmark gene sets in the intestinal SC cluster of irradiated WT mice compared with *Myc^{Δ2-540/Δ2-540}* mice (also see Table S2). **(B)** Deletion of the *Myc^{2-540}* region causes a modest decrease in conserved MYC target (Zielke et al, 2022) expression, whereas after irradiation, strong induction of MYC targets is seen in WT compared with *Myc^{Δ2-540/Δ2-540}* mice. Top 5 differentially induced MYC targets in the stem cell compartment are marked (also see Table S3). **(C)** Targets repressed by MYC (Diamant et al, 2025) are not affected to the same extent as conserved MYC targets after radiation. **(D)** Altered regulation of MYC and mTORC target *Eif4ebp1* (4EBP1) (D) scRNA-seq data (2 dpi). **(E)** IHC showing the amount of 4EBP1 and phosphorylated 4EBP1 (phospho-4EBP1) before and after radiation. **(F)** Quantification of the IHC staining in E using QuPath version 1. Each dot represents an individual annotated region that was measured (sections from at least two mice per category were analyzed). In (B, C), red lines denote the median expression of MYC targets and blue dots mark targets with significant (*P* < 0.05) log_2 fold changes between WT and *Myc^{Δ2-540/Δ2-540}*. **(B, C)** Wilcoxon's test was used for statistical analysis in (B, C). Statistical significance in F was calculated using an unpaired *t* test (two-tailed), and corresponding *P*-values are shown. A nonsignificant statistical difference is marked by ns. Error bars in F denote the mean ± SD. Magnification bars in E = 50 μm. SC: stem cell; TA: transit amplifying cell; EC: enterocyte.

the stem, transit amplifying, and enterocyte progenitor cell clusters, confirming their stem cell potential (Fig 5D). In contrast, the *Myc^{Δ2-540/Δ2-540}* mice almost completely lacked *Ly6a* (Sca1) expression

after irradiation (Fig 5D). Furthermore, the failure of *Myc^{Δ2-540/Δ2-540}* mice to effectively mobilize *Ly6a* (Sca1)⁺ cells after irradiation could be attributed to reduced proliferation linked to a loss of MYC target

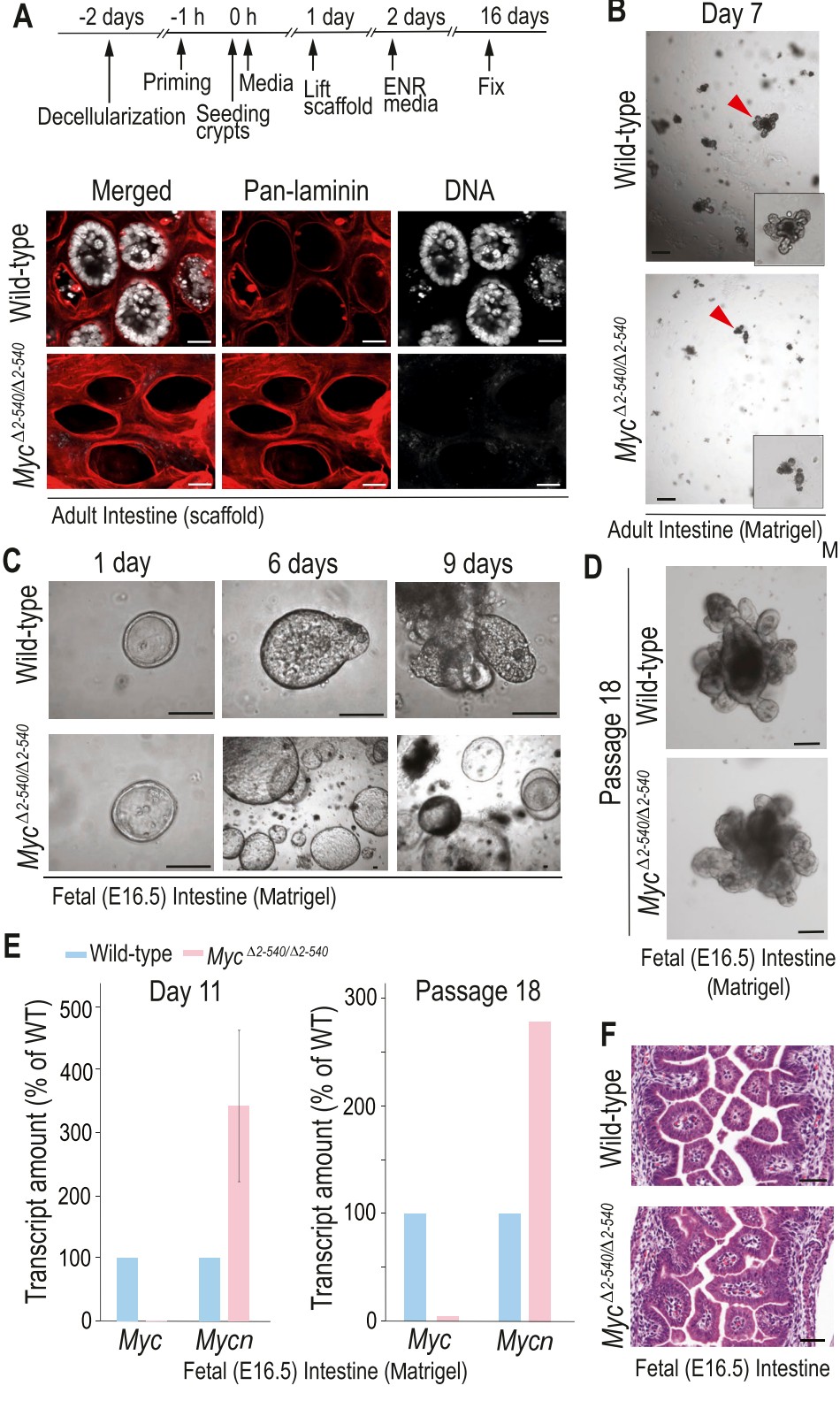

Figure 4. **Distinct MYC requirements in adult and fetal intestinal tissues during regeneration in vitro.**
**(A)** Immunofluorescence analysis showing failure of adult $Myc^{\Delta2\text{-}540/\Delta2\text{-}540}$ crypts to re-epithelialize intestinal scaffolds. Representative images from two independent experiments are shown. **(B)** Crypts from $Myc^{\Delta2\text{-}540/\Delta2\text{-}540}$ adult intestine fail to establish 3D organoid cultures in Matrigel. Representative images from at least three independent experiments are shown. **(C)** Intestinal epithelium from $Myc^{\Delta2\text{-}540/\Delta2\text{-}540}$ E16.5 embryos can initiate spheroid formation in Matrigel but fail to generate budding organoids with mature crypt-like structures reminiscent of adult cultures during 10 d of culture. Representative images from three independent experiments are shown. **(D)** Late-passage organoids showing recovery of the budding phenotype in $Myc^{\Delta2\text{-}540/\Delta2\text{-}540}$ fetal intestinal 3D cultures over time (n = 1 experiment for each category). **(E)** qPCR analysis showing the almost complete absence of *Myc* expression in $Myc^{\Delta2\text{-}540/\Delta2\text{-}540}$ spheroids and potential compensation by N-MYC in 8- to 9-d-old 3D cultures (WT: n = 2; $Myc^{\Delta2\text{-}540/\Delta2\text{-}540}$: n = 2) and in organoids after passage 18 (n = 1 for each category). **(F)** $Myc^{\Delta2\text{-}540/\Delta2\text{-}540}$ E16.5 fetal intestine is morphologically similar to WT in vivo. H&E-stained sections are shown. Magnification bars in (A, D) = 20 μm, (B, F) = 50 μm, (C) = 4 μm. Error bars in (E) denote the mean ± SD. Red arrowheads in (B) denote organoids magnified in the inset.

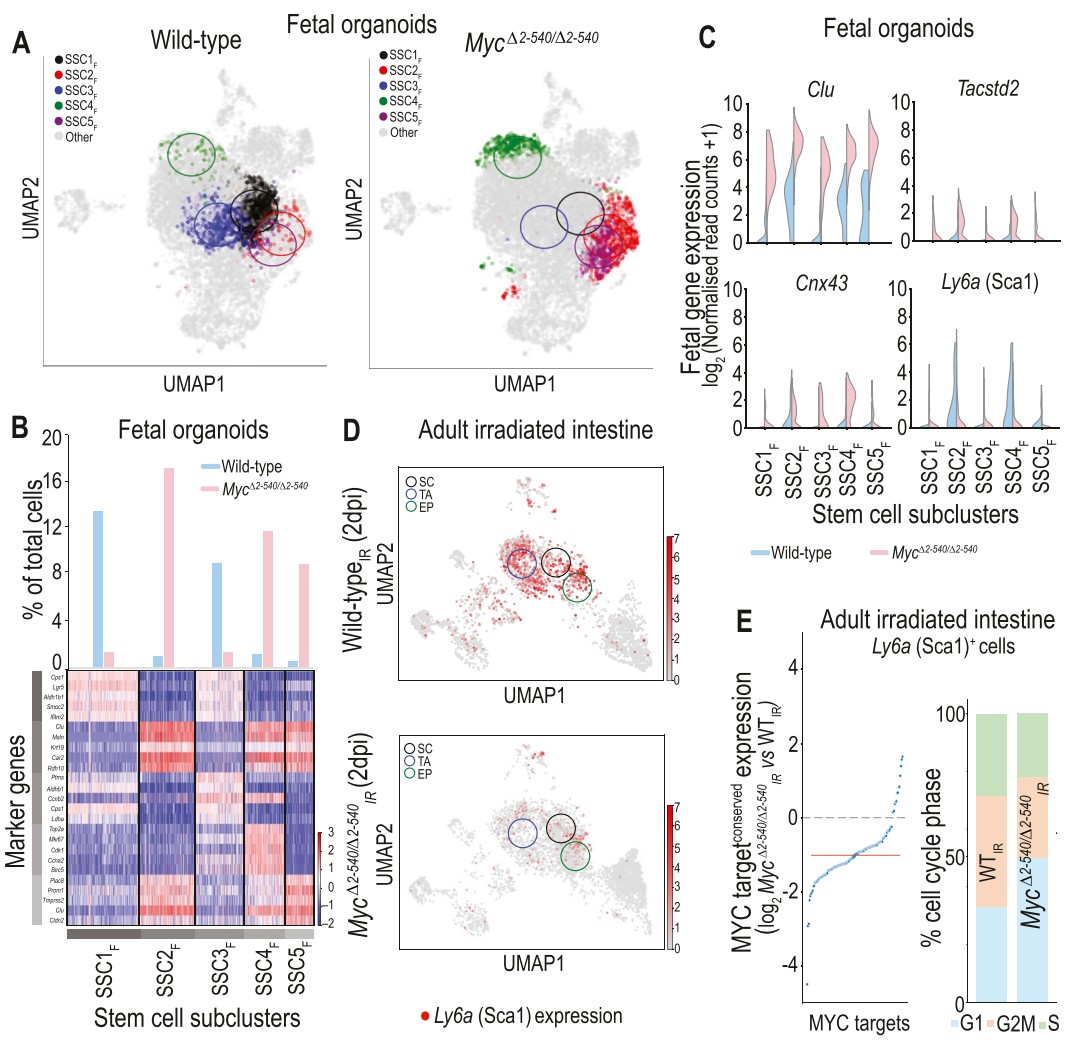

**Figure 5. Mobilization of *Ly6a* (Sca1)+ cells during intestinal regeneration in in vitro 3D organoid cultures and in vivo after irradiation damage is *Myc²⁻⁵⁴⁰*-dependent.**
**(A)** UMAP plots showing the distribution of five stem cell subclusters identified in WT and *Myc^Δ2-540/Δ2-540* fetal intestinal 3D cultures. The mean value of x and y for all the cells belonging to that subcluster was used to determine the center and marked with a circle on the UMAP (fixed radius of 1.5). **(B)** Proportion of stem cell subclusters relative to the total number of cells from each genotype in fetal intestinal 3D cultures (top) and heat map showing the expression of de novo identified marker genes for each stem cell subcluster (bottom). **(C)** *Myc^Δ2-540/Δ2-540* fetal intestinal cultures express high levels of fetal/regeneration-associated markers but have much reduced levels of *Ly6a* (Sca1) compared with WT. **(D)** UMAP analysis of scRNA-seq data showing the loss of *Ly6a* (Sca1) induction in *Myc^Δ2-540/Δ2-540* mice compared with WT after irradiation. *Ly6a* (Sca1)-expressing cells (red) are distributed in SC, TA, and EP populations. Clusters were marked in the same way as in (A). **(E)** scRNA-seq analysis showing the reduced expression of conserved MYC targets and altered cell cycle distribution of *Ly6a* (Sca1)+ cells from irradiated *Myc_IR^Δ2-540/Δ2-540* intestine compared with irradiated WT intestine (WT_IR) (2 dpi). The red line in E denotes the median expression of conserved MYC targets. SC, stem cell; TA, transit amplifying cell; EP, enterocyte progenitor.
Source data are available for this figure.

induction (Fig 5E). These results demonstrate that although fetal intestinal 3D cultures can be propagated in almost complete absence of MYC, the adult repair and regenerative state is critically coupled to a MYC-dependent proliferation and mobilization of regenerative stem/progenitor cells.

## Tumors also show *Myc²⁻⁵⁴⁰*-dependent *Ly6a* (Sca1)+ cell recruitment

The mechanism of repair/regeneration is tightly coupled to development of neoplastic lesions; we therefore investigated whether defective recruitment of *Ly6a* (Sca1)-expressing cells could explain the tumor resistance of *Myc^Δ2-540/Δ2-540* mice. We have earlier shown that *Myc^Δ2-540/Δ2-540* mice and mice deficient of a single enhancer at the *Myc* locus (*Myc-335⁻/⁻*) display 98% and 70% resistance to intestinal polyps on *Apc^Min* background (Sur et al, 2012; Dave et al, 2017). Analysis of disease-free survival showed that *Apc^Min/+*; *Myc-335⁻/⁻* mice survived longer than WT mice, but eventually had to be euthanized by 1 yr of age because of intestinal polyposis (median survival = 306 d [n = 7] compared with 156 d for *Apc^Min/+* [n = 9]). In contrast, most (94%) of *Apc^Min/+*; *Myc^Δ2-540/Δ2-540* mice lived without observable adverse health effects for more than 500 d, but developed symptoms consistent with intestinal tumorigenesis (e.g., weight loss, blood in stool, rectal prolapse, and/or abdominal

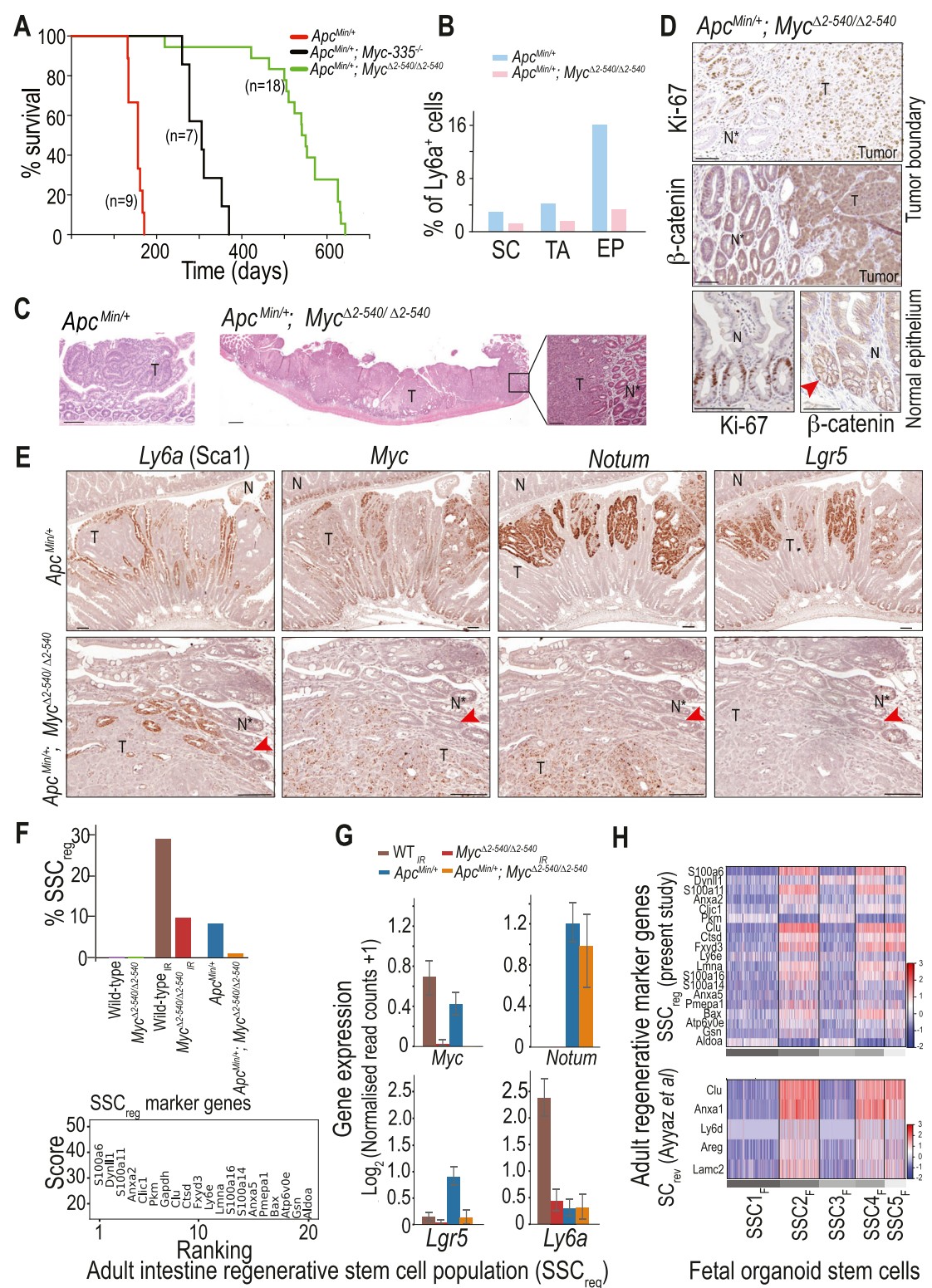

**Figure 6. Tumors require *Myc²⁻⁵⁴⁰*-dependent *Ly6a* (Sca1)⁺ cell mobilization but differ from wound repair with respect to *Notum* expression.**

**(A)** *Apc^Min/+^; Myc^Δ2-540/Δ2-540^* mice survive to old age (average = 18.2 mo, n = 18) compared with *Apc^Min/+^* (average = 5 mo, n = 9) because of their inherent tumor resistance. **(B)** Tumor resistance of *Apc^Min/+^; Myc^Δ2-540/Δ2-540^* mice is correlated with a decrease in *Ly6a* (Sca1)-expressing cells within the stem cell (SC), transit amplifying cell (TA), and enterocyte progenitor (EP) cell populations as determined by scRNA-seq analysis. **(C)** Polypoid nature of *Apc^Min/+^* tumors (4 mo of age) is predominantly lost in the few tumors that develop in old *Apc^Min/+^; Myc^Δ2-540/Δ2-540^* mice. A tumor from 1.5-yr-old *Apc^Min/+^; Myc^Δ2-540/Δ2-540^* is shown. The magnified region shows a well-demarcated boundary between the tumor and the normal epithelium. **(D)** IHC showing proliferation (Ki-67) and active nuclear *β*-catenin within the tumor of aging (1.5 yr old) *Apc^Min/+^;*

bloating) before 2 yr of age. The median survival was 546 d (18.2 mo), corresponding to normal old age in the C57BL/6 strain (18–24 mo) (Fox et al, 2006) (Fig 6A). To further investigate the mechanism of tumor resistance conferred by $Myc^{2-540}$ loss, we performed scRNA-seq analysis on intestines from 4-mo-old $Apc^{Min/+}$ and $Apc^{Min/+};$ $Myc^{\Delta2-540/\Delta2-540}$ mice. At this time point, $Apc^{Min/+}$ intestines have on an average ~50 polyps, whereas $Apc^{Min/+}; Myc^{\Delta2-540/\Delta2-540}$ intestines are almost completely devoid of tumors (Dave et al, 2017). Similar to our results from irradiation experiments, the $Apc^{Min/+}; Myc^{\Delta2-540/\Delta2-540}$ mice had a lower number of $Ly6a$ (Sca1)$^+$ cells compared with the $Apc^{Min/+}$ intestines (Fig 6B). To determine the role of $Ly6a$ (Sca1)$^+$ cells in tumorigenesis, we characterized the late tumors that do develop in very old $Apc^{Min/+}; Myc^{\Delta2-540/\Delta2-540}$ mice. These tumors are often relatively large, but have a nonpolypoid morphology that is very distinct from polyps that form in $Apc^{Min/+}$ mice (Fig 6C). $Apc^{Min/+}; Myc^{\Delta2-540/\Delta2-540}$ tumors often have invaded radially to encircle the entire small intestine, and display a lateral "pushing edge" growth pattern, instead of the finger-like pattern common to polyps. The tumors were proliferative and Wnt-dependent based on the expression of Ki-67 and nuclear $\beta$-catenin, respectively (Fig 6D). These results establish that although MYC upstream super-enhancer region is necessary for formation of a particularly common type of intestinal tumors, loss of this element does not completely eliminate tumorigenesis in the intestine. The phenotype of $Myc^{2-540}$ deletion is thus consistent with disruption of tissue-specific regulatory mechanisms rather than a blanket block of $Myc$ expression. This is further corroborated by the observation that $Myc^{\Delta2-540/\Delta2-540}$ mice do not show resistance to lymphoid proliferative lesions/tumors induced by PTEN loss (Fig S6A and B) as opposed to the striking resistance for intestinal tumors.

To further characterize the intestinal tumors in aged $Apc^{Min/+}; Myc^{\Delta2-540/\Delta2-540}$ mice, we performed ISH on tumors from old $Apc^{Min/+}; Myc^{\Delta2-540/\Delta2-540}$ and from 4-mo-old $Apc^{Min/+}$ mice.

We used ISH probes against $Myc$ and $Lgr5$ to determine the contribution of $Myc$ and adult stem cells to tumorigenesis in $Apc^{Min/+}$ and $Apc^{Min/+}; Myc^{\Delta2-540/\Delta2-540}$ mice. Because $Ly6a$ (Sca1) was altered in $Myc^{\Delta2-540/\Delta2-540}$ intestine both during in vivo and in vitro regeneration, we also studied its expression in the tumors. Finally, $Notum$, which provides a growth advantage to $Apc^{-/-}$ stem cells over the neighboring WT counterparts, was selected to probe $Apc$ mutant cells in and around the tumors (Flanagan et al, 2021; Yan et al, 2021; Yum et al, 2021). Most cells of the normal epithelium of both $Apc^{Min/+}$ and $Apc^{Min/+}; Myc^{\Delta2-540/\Delta2-540}$ did not express $Notum$ or $Ly6a$ (Sca1). However, unlike most of the normal epithelium, cells both within and close to the tumor from both $Apc^{Min/+}$ and $Apc^{Min/+}; Myc^{\Delta2-540/\Delta2-540}$ commonly expressed $Ly6a$ (Sca1) (Fig 6E). Based on morphology, the $Ly6a$ (Sca1)$^+$ cells within tumors appeared to represent both tumor

cells and residual normal cells trapped within the tumor. The tumor also appeared to induce $Ly6a$ (Sca1)$^+$ expression in normal cells that were located at the boundary between malignant and normal epithelium, suggesting that the tumor is capable of damaging surrounding normal tissue and inducing a wound-healing response. Normal-looking crypts expressing $Ly6a$ (Sca1) at the tumor boundary did not express detectable transcripts of $Myc$, or $Notum$, which is a specific marker of $Apc^{-/-}$ cells (Figs 6E and S7), suggesting that these cells are at a growth disadvantage relative to the $Myc$-expressing tumor cells (Johnston, 2014; Di Giacomo et al, 2017). The expression pattern also indicates that the expression of $Ly6a$ (Sca1) is MYC-independent. Unlike the normal epithelium, tumors in $Apc^{Min/+}; Myc^{\Delta2-540/\Delta2-540}$ expressed $Myc$ potentially via additional regulatory regions that are still intact in the $Myc^{\Delta2-540/\Delta2-540}$ mice. Single-cell analysis of the stem cell population from $Apc^{Min/+}$ tumor-bearing epithelium and WT irradiated intestine showed that both contained a population of stem cells (SSC$_{reg}$) enriched in fetal/regenerative markers like $Clusterin$ ($Clu$) (Fig 6F).

To determine how the wound and cancer-associated SSC$_{reg}$ population of stem cells in the adult relate to the fetal stem cells described earlier, we checked the expression of SSC$_{reg}$ marker genes (Fig 6F) in stem cell subclusters from fetal 3D organoid cultures. The adult regenerative gene set was more highly expressed in the fetal SSC2$_F$, SSC4$_F$, and SSC5$_F$ subclusters, which are enriched in $Myc^{\Delta2-540/\Delta2-540}$ cultures compared with the same cells of the WT mice (Fig 6H). This likely reflects the fetal reversion of adult stem cells during regeneration and cancer. However, unlike the fetal state that can support growth even with absent or reduced $Myc$ and $Ly6a$ (Sca1) expression, the adult regenerative state appears to critically depend on $Myc$- and $Ly6a$ (Sca1)-expressing cells. In addition, SSC$_{reg}$ population from $Apc^{Min/+}$ tumor–bearing epithelium and WT irradiated intestine could be distinguished from each other based on $Notum$ expression. $Notum$ was expressed exclusively in an $Apc^{Min/+}$ background, and its expression was not affected by $Myc^{2-540}$ deletion (Fig 6G). Because $Notum$ in $Apc^{-/-}$ cells is required for inhibition of neighboring WT stem cells and expansion of tumor cells, it would explain why wound-associated proliferation stops, whereas tumors continue to proliferate and invade into the surrounding tissue.

## Discussion

Our results presented here establish the long-sought mechanistic link between wound healing and cancer, and show that rapid

---

$Myc^{\Delta2-540/\Delta2-540}$ mice. The bottom-most panel shows Ki-67 and active nuclear $\beta$-catenin within the normal epithelium of aging $Apc^{Min/+}; Myc^{\Delta2-540/\Delta2-540}$ intestine. **(E)** ISH showing the expression of $Ly6a$ (Sca1), $Myc$, $Notum$, and $Lgr5$ in tumors from 4-mo-old $Apc^{Min/+}$ and 1.5-yr-old $Apc^{Min/+}; Myc^{\Delta2-540/\Delta2-540}$ mice. **(F)** Top panel: percentage of cells belonging to the regenerative stem cell subcluster (SSC$_{reg}$), common to both WT irradiated (WT$_{IR}$; 2 dpi) and $Apc^{Min/+}$ intestines, is shown. Percentage is relative to the total stem cell number. Bottom panel: de novo identified marker genes of SSC$_{reg}$ population. **(G)** Expression of $Myc$, $Notum$, $Lgr5$, and $Ly6a$ (Sca1) in the stem cell subclusters across different genotypes and conditions. **(H)** Heat map showing the expression of (top panel) SSC$_{reg}$ marker genes (from (F)) and (bottom panel) revival stem cell (SC$_{rev}$) markers (Ayyaz et al, 2019) across the stem cell subclusters identified in fetal (E16.5) organoid cultures from WT and $Myc^{\Delta2-540/\Delta2-540}$ mice (SSC1$_F$–SSC5$_F$). The $Gapdh$ gene from the SSC$_{reg}$ marker list was not detected in scRNA-seq data from fetal organoid cultures. Magnification bars = 100 $\mu$m except in (C) where the magnification bar for an overview image of the $Apc^{Min/+}; Myc^{\Delta2-540/\Delta2-540}$ tumor = 500 $\mu$m. T marks the tumor region, N* denotes normal epithelium bordering the tumor, and N denotes normal epithelium away from the tumor. The arrowhead in (B) marks the cell with nuclear $\beta$-catenin staining in normal epithelium. The arrowhead in (E) marks one $Lgr5$-expressing crypt across panels. SC, stem cell; TA, transit amplifying cell; EP, enterocyte progenitor.
Source data are available for this figure.

proliferation during tissue regeneration and cancer uses *MYC* regulatory mechanisms that are genetically distinct from those used during normal homeostasis. Although the role of MYC in regeneration and cancer is well established (Ashton et al, 2010; Kim et al, 2018; Sodir et al, 2022; Mule et al, 2024), our work expands these studies further by providing a function to a large upstream super-enhancer region of *MYC*. This region is highly conserved and carries the highest cancer susceptibility burden in our entire genome. Our prior study (Dave et al, 2017) showed that the entire region can be deleted from mice without severe phenotypic consequence, making the existence and conservation of such a cancer-predisposition region enigmatic. Here, we solve this mystery, showing that the MYC super-enhancer region is normally required for regeneration of intestinal tissues after damage. The conservation of this region can thus be explained by the critical role that intestinal health has on survival of infants and young mammals.

We find here also that organoids derived from mice lacking the MYC super-enhancer region completely fail to grow in culture. This indicates that cell culture models that are often considered to resemble normal in vivo tissue renewal actually drive cells to adopt an exponential stem cell proliferation program associated with regeneration and tumorigenesis. In striking contrast to the adult organoids, embryonic intestinal cells derived from MYC super-enhancer region–deficient mice could be cultured as organoids, suggesting that some fetal regenerative mechanisms are shut down in the adult state, making adult tissue regeneration completely dependent on MYC.

Given that up-regulation of ribosome biogenesis is required for MYC-driven proliferation (van Riggelen et al, 2010; Pihlajamaa et al, 2022; Zielke et al, 2022), the simplest model consistent with our findings is that different levels of protein synthesis are needed for distinct proliferative cell fate choices. In this model, relatively low levels of protein synthesis and MYC are sufficient for proliferation that maintains stem cell number in tissue renewal, whereas processes that require an exponential increase in the number of somatic stem cells, such as cell proliferation in vitro, regeneration, and tumorigenesis (Fig 7), require high levels of MYC and protein synthesis. Interestingly, the model has similarities to growth control in unicellular species, where nutrient availability specifies the protein synthetic rate and controls the balance between non-proliferative and proliferative states (Jorgensen & Tyers, 2004). Multicellularity could have evolved simply by the addition of a hierarchy of stem cells with new proliferative fates, which remain controlled by the same input variable, the total metabolic activity per genome. Consistently with this model, ES cells lacking all forms of MYC enter into a diapause-like nonproliferative state in vitro (Scognamiglio et al, 2016), and we also observe here that adult intestinal cells from $Myc^{\Delta2-540/\Delta2-540}$ mice enter a nonproliferative state under full mitogen stimulation (organoid culture).

Our work has important consequences for drug development. In particular, compounds that inhibit proliferation of normal cells in culture have commonly been discarded during the development process because of toxicity, even when a subset of them may actually target the very mechanism that is distinct between normal proliferative cells and cancer cells in vivo. Our finding that the $Myc^{2-540}$ element and MYC activity are required for adult but not fetal organoid growth provides an effective means to identify drugs that specifically target MYC, paving a way for development of a novel

class of effective chemopreventive and antineoplastic agents. Furthermore, given that the first successful phase 1 clinical trial for MYC inhibitor (OMO-103) was recently published (Garralda et al, 2024), our results suggest a potential impact for combined therapies for cancer that include radiation and MYC inhibition.

# Materials and Methods

### Animals

Mouse lines used in this study have been described previously (Dave et al, 2017). The conditional knockout mice were generated on a C57BL/6N genetic background. After Cre-mediated deletion of the 538-kb region, mice were backcrossed to C57BL/6J for two generations and subsequently intercrossed. All mice were kept on a 12:12-h light cycle in standard housing conditions and provided with food (Teklad Global Diet 2918; Envigo) and water ad libitum. Animal experiments were done in accordance with the regulations and with approval of the local ethical committee (Stockholms djurförsöksetiska nämnd). Mice were irradiated with Gammacell 40 Exactor that uses cesium-137 as a source for $\gamma$-rays. Mice were given two doses of 6 Gy with a 6-h interval in between the doses. After irradiation, mice were provided with soft food. For measuring survival of mice with $Apc^{Min}$ mutation (strain 002020 from The Jackson Laboratory), the general health of the animal was monitored and mice were euthanized when they reached the humane endpoint according to the ethical permit. Mice that had to be euthanized for reasons unrelated to intestinal tumorigenesis were not included in the analysis. For PTEN mutation (strain 01XH3 from NCI mouse repository), mice were euthanized at the humane endpoint, predominantly because of lymphoproliferative lesions that manifested as nodules/lumps on the body of mice.

### Immunohistochemistry and RNA ISH

PFA-fixed and paraffin-embedded (FFPE) tissues were sectioned at 5 $\mu$m thickness and subjected to immunohistochemistry using standard methods. For antigen retrieval, the sections were boiled in microwave for 20 min in 10 mM citrate buffer, pH 6.0, and subsequently processed for respective antibody stainings. The following antibodies and dilutions were used: anti-Ki-67 (1:200; Abcam), anti-lysozyme (1:500; Abcam), anti-Muc2 (1:500; Abcam), anti-Chga (1:1,000; Abcam), anti-4EBP1 (1:100; Abcam), and anti-phospho-4EBP1 (1:2,500; Cell Signaling). ISH was also performed on 5-$\mu$m-thick sections using the RNAscope technology and according to the manufacturer's protocol. Probes against mouse *Myc* (catalog nr: 413451), *L-Myc* (catalog nr: 552711), *Lgr5* (catalog nr: 312171), *Clu* (catalog nr: 427891), *Ly6a* (Sca1) (catalog nr: 427571), and *Notum* (catalog nr: 428981) were used. RNA integrity was confirmed using the control probe for *Ubc* (catalog nr: 310771).

### Quantification of microscopy data

Longitudinally sectioned crypt–villus structures were used for quantification of crypt depth, villus height, Ki-67, and cell

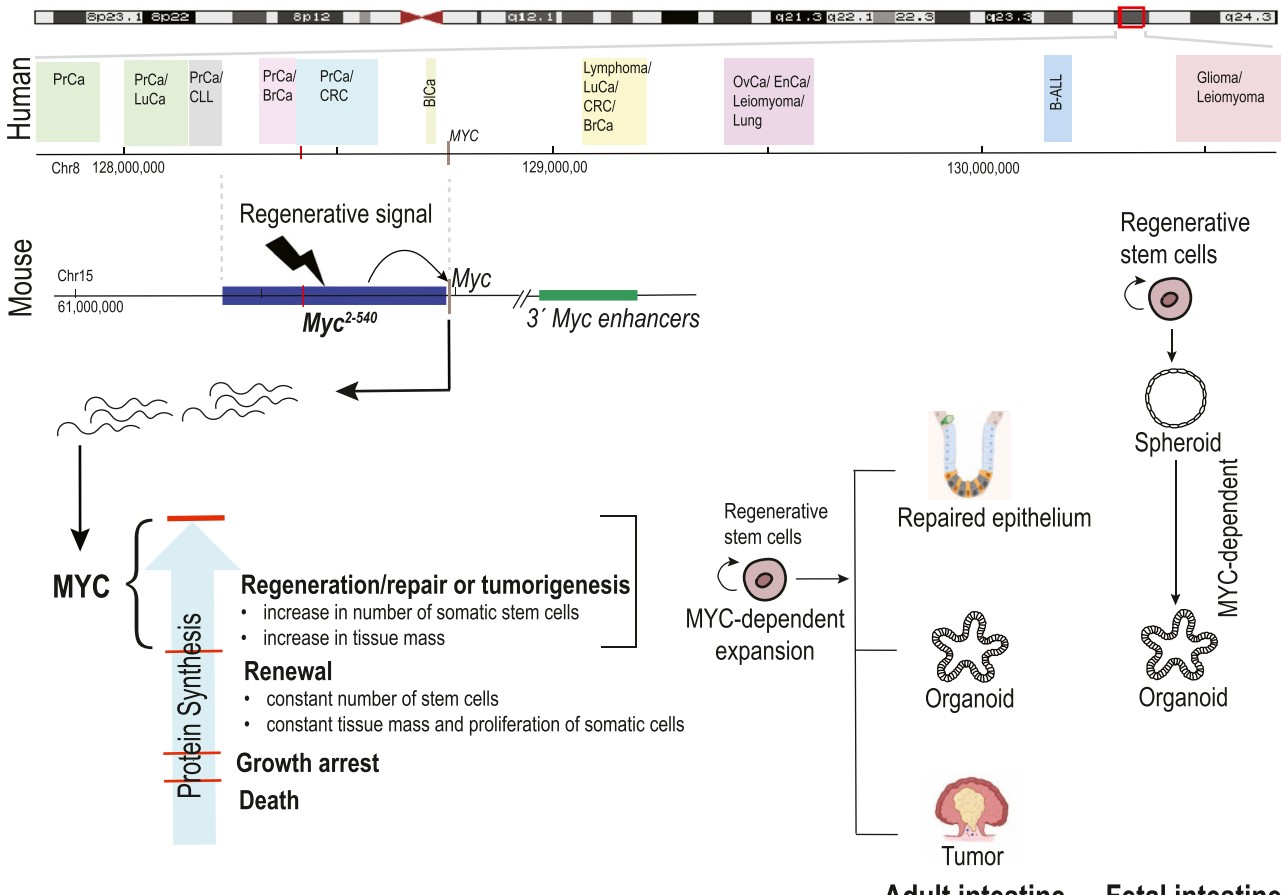

**Figure 7. Model describing how cancer susceptibility-associated super-enhancer region *Myc*[2-540] drives regenerative growth in the fetal and adult intestine and in cancer.**
Because most of the MYC targets are proteins involved in ribosomal biogenesis, the model further describes how different levels of protein synthesis can result in different proliferative cell fate choices ranging from normal homeostasis to regeneration and cancer. Schematic representation of the cancer risk distribution within the gene desert 1 Mb upstream and 2 Mb downstream of the *MYC* oncogene on human chromosome 8q24 is shown. In the topmost is shown human chromosome 8, with the 3-Mb 8q24 region outlined in red. The magnified view of the 3-MB region is shown below, with colored boxes showing risk regions for prostate cancer (PrCa), chronic lymphocytic leukemia (CLL), breast cancer (BrCa), colorectal cancer (CRC), bladder cancer (BlCa), lymphoma, lung cancer (LuCa), ovarian cancer (OvCa), endometrial cancer (EnCa), leiomyoma, acute lymphoblastic leukemia (B-ALL), and glioma. A red vertical line crossing the human genomic coordinate axis marks the enhancer element *MYC-335* that carries the CRC risk–associated SNP rs6983267 (Sur et al, 2012). GRCh37/hg19 human and GRCm38/mm10 genome coordinates are shown. The schematic of the cancer susceptibility region is based on GWAS catalog data (https://www.ebi.ac.uk/gwas/). The syntenic *Myc2-540* region located upstream of *Myc* on mouse chromosome 15 and deleted from the mouse genome in this study is shown in blue on the schematic for mouse chromosome 15.

type–specific marker stainings (CHGA, MUC2, LYZ). For crypt depth, the distance from the bottom of the crypt to villus junction was measured in microns using QuPath. Similarly, villus height was measured from the crypt–villus junction to the villus tip. Ki-67⁺, CHGA⁺, MUC2⁺, and LYZ⁺ cells were manually counted. For quantifying 4EBP1 and phospho-4EBP1 stainings, sections were manually annotated in QuPath, with each an-notated region containing several crypt–villus structures. Subsequently, the % DAB-positive pixels were quantified per annotated region with QuPath. For quantification of *Mycl* ISH stainings, positive punctate stainings (dots) within longitudi-nally cut crypts were manually counted under a microscope.

## qRT–PCR

Approximately 1 cm of the mouse distal ileum was cut into small pieces and stored in RNAlater/RNAProtect (QIAGEN). For 3D cultures, plates were placed on ice to melt the Matrigel and organoids/spheroids were collected by centrifugation. RNA was prepared using a combination of TRIzol (Ambion) and RNeasy Mini Kit (QIAGEN). cDNA was synthesized from either 1 μg (tissue) or 100 ng (organoid) of total RNA using High Capacity cDNA Reverse Transcription Kit (Applied Biosystems). qPCR was performed using SYBR Green Master Mix (Thermo Fisher Scientific) on a LightCycler 480 instrument (Roche). Primers for *Myc* and *Actin* were as described previously (Dave et al,

2017). For *Mycn* PrimeTime qPCR, primers from Integrated DNA Technologies (IDT) were used (Mm.PT.58.29713369).

### scRNA-seq

Mouse intestinal crypts were isolated according to Sato and Clevers (Sato & Clevers, 2013) and dissociated with TrypLE Express (37°C/20 min; Invitrogen), and single-cell suspensions were run on a 10x Chromium controller using the Chromium Next GEM Single Cell 3′ kit v3.1 (catalog nr: PN-1000268) and according to the manufacturer's protocol (10x Genomics). The cell number targeted was 6000. For nonirradiated samples, crypts were isolated from ileum and single-cell suspensions of FACS-sorted 7AAD⁻ (Life Technologies), CD45⁻ (eBioscience), CD31⁻ (eBioscience), TER-119⁻ (eBioscience), and EpCAM⁺ (eBioscience) cells were used. For irradiated samples, crypts from the whole small intestine were isolated and 7AAD⁻ FACS-sorted cells were used. For single-cell suspensions from 3D cultures, organoids were treated with TrypLE for 20 min and cells were fixed with PFA according to 10x Next GEM Single Cell Fixed RNA Sample Preparation Kit (catalog nr: 1000414). Libraries from fixed cells were prepared using Next GEM Single Cell Fixed RNA Hybridization and Library Kit (catalog nr: 1000415) and according to the manufacturer's protocol (10x Genomics). All libraries were sequenced on Illumina sequencing platform NovaSeq 6000 or NextSeq 2000. Sequencing data were processed using either Cell Ranger (v6.0.1) (10x Genomics) or Cell Ranger (v7.2.0) (for fixed cells with Cell Ranger multi-pipeline) (10x Genomics). Quality control (QC) analysis of the Cell Ranger output was done in Scanpy (v1.9.1) (Wolf et al, 2018) as follows: mean and SD of genes in all cells (Mean [$n_{genes}$], SD [$n_{genes}$]) and mean, median, median absolute deviation (MAD), and SD of total counts in all cells were calculated. Cells were kept with the number of expressed genes larger than (median [$n_{genes}$]-MAD [$n_{genes}$]) total read counts in the range of (Mean [$n_{counts}$]-1.5*MAD [$n_{counts}$], Mean [$n_{counts}$]+3*SD [$n_{counts}$] and percentage of expressed mitochondrial genes larger than its median+3*MAD). In addition, genes expressed in less than 0.1% of cells were filtered out. The final number of cells used for further analysis was as follows: WT: 8509 cells, *Apc*$^{Min/+}$; *Myc* $^{\Delta 2-540/\Delta 2-540}$: 13763 cells, *Apc*$^{Min/+}$: 7468 cells, *Myc* $^{\Delta 2-540/\Delta 2-540}$: 5809 cells, *Myc* $^{\Delta 2-540/\Delta 2-540}$ irradiation day 2: 4,863 cells and 3,102 cells from WT irradiation day 2. The number of cells analyzed from fixed samples (3D organoid cultures) was as follows: 6,699 cells from WT and 3,879 cells from *Myc* $^{\Delta 2-540/\Delta 2-540}$. Subsequently for dimensional reduction, the top 2,000 highly variable genes were selected based on scRNA raw read count matrix (scanpy.pp.highly_variable_genes; flavor = 'seurat_v3'). There were no mitochondrial genes in the top 2,000 highly variable genes. The raw read count matrix with the top 2,000 highly variable genes was subjected to the Bayesian variational inference model scVI (v0.16.4) (Lopez et al, 2018) to infer a latent space for all single cells. The batch effect among different samples was automatically removed for all samples within the scVI. After that, two-dimensional visualizations were calculated using Uniform Manifold Approximation and Projection (UMAP). Clustering analysis was performed using the Leiden algorithm. First, a k-nearest–neighbor graph from the scVI latent space was constructed, and then, the sc.tl.leiden function in Scanpy was used with the resolution of 1. Clusters were annotated into intestinal cell types based on published marker genes (Yum et al, 2021). Sub-cluster analysis was done within SC clusters with a resolution set to 0.3 (3D organoid cultures) or 0.2 (adult intestinal epithelium). The location of the cell-type clusters/subclusters was marked on the UMAP by calculating the mean value of x and y of all the cells belonging to that cluster/subcluster to find the center and then circling it on the UMAP (fixed radius of 1.5). For further analysis, the single-cell raw read count matrix of all the genes was input into R package Scran (v1.24.1) to calculate the size factor for each cell (Lun et al, 2016). To normalize the gene expression across cells, each gene's read count in a cell was divided by that cell's size factor in Scanpy. The normalized gene read counts were log₂-transformed, where 1 pseudocount was added to gene's read count to avoid a negative infinity value in the analysis (log₁ *P*-value). The z-score for the log₁ *P*-value for each gene was calculated and used for heat map visualization of cell type–specific marker genes. Differentially expressed genes between the genotypes were determined using Scanpy's function sc.tl.rank_genes_groups. The analysis was done specifically in the stem cell (SC), transit amplifying cell (TA), and enterocyte (EC) clusters. Within this function, the Wilcoxon test was used to calculate *P*-values and the false discovery rate by applying the Benjamini–Hochberg approach. The log fold changes of the 126 known MYC targets (Zielke et al, 2022) were retrieved from the differential gene expression results and plotted using a dot plot. Polr1f was not detected in our dataset and was excluded from analysis (Table S3). The ranking of the genes on the x-axis was based on the median log fold change values of each MYC target gene among stem cell (SC), transit amplifying cell (TA), and enterocyte (EC) clusters combined. Gene set enrichment analysis was done using the R package clusterProfiler (v4.4.4). Gene signatures were obtained from Gene Ontology Biological Process gene sets (Ashburner et al, 2000) and the MSigDB "Hallmark gene sets" (Liberzon et al, 2015). All the expressed genes in the SC cluster were pre-ranked by their log fold changes, which were determined as described in the previous section. GSEA was then performed using ClusterProfiler (ClusterProfiler::GSEA; pvalueCutoff = 1, pAdjustMethod = ´BH´). The list of GSEA results is provided in Table S2.

### Single-cell ATAC-seq (scATAC-seq)

scATAC-seq was performed according to Cheng et al (2021) *Preprint*. Single-cell suspensions of mouse intestinal crypts were prepared as in Andrews et al (2021), and single cells were FACS-sorted into a 384-well plate with 3 μl lysis buffer in each well (0.03 ml 1 M Tris, pH 7.4, 0.0078 μl 5 M NaCl, 0.075 μl 10% IGEPAL, 0.075 μl RNase inhibitor, 0.075 μl 1:1.2 M ERCC, 2.7372 μl H₂O). After cell lysis, 1 μl of the lysis buffer containing the nuclei was used for scATAC library preparation, whereas the remaining 2 μl was discarded. The scATAC in situ tagmentation was performed with 2 μl of the Tn5 tagmentation mix (0.06 μl 1 M Tris, pH 8.0, 0.0405 μl 1 M MgCl₂, 0.2 μl Tn5) at 37°C for 30 min. After tagmentation, 2 μl of the supernatant was aspirated and the nucleus was then washed once with 10 μl ice-cold washing buffer (0.1 μl 1 M Tris, pH 7.4, 0.02 μl 5 M NaCl, 0.03 μl 1 M MgCl₂, 9.85 μl H₂O). The remaining Tn5 was inactivated by adding 2 μl 0.2% SDS/ Triton X-100 and incubating at room temperature for 15 min followed by 55°C for 10 min. Before barcoding PCR, genomic DNA was extracted from chromatin by adding 2 μl Proteinase K (0.0107 au/ml)

and incubation at 55°C for 1 h followed by heat inactivation at 70°C for 15 min. Barcoding PCR was done using KAPA HiFi PCR Kit (Roche) in a final volume of 25 $\mu$l. The PCR condition was 72°C/15 min, 95°C/45 s, (98°C/15 s, 67°C/30 s, 72°C/1 min) × 22 cycles, 72°C/5 min, and then 4°C hold. After the PCR, 2 $\mu$l reaction from each well was pooled and cleaned up twice using SPRI beads (at 1.3X volume) and the library was sequenced on Illumina NextSeq 550 as paired-end, dual index (37 + 37+8 + 8). For scATAC-seq data, raw sequence reads were trimmed based on quality (phred-scaled value of >20) and the presence of Illumina adapters, and then aligned to the mm10 genome build using BWA (Li & Durbin, 2010). Reads that were not mapped, not with primary alignment, missing a mate, mapq <10, or overlapping ENCODE blacklist regions (Amemiya et al, 2019) were removed. A custom script was used to summarize the paired-end reads into a de-duplicated fragment file suitable for downstream analysis. scATAC-seq data were analyzed using ArchR. Cells with fewer than 1,000 ATAC-seq reads mapping to the transcription start site, or with TSS enrichment <4, were excluded from analysis. The "iterative LSI" function with maxClusters = 6 was used for cluster analysis. Clusters with the high expression of "*Muc2*" were annotated as goblet cells and discarded. Clusters high in "*Ascl2*" and "*Lgr5*" were annotated as stem cells, whereas remaining intermediate/low-expressing clusters were annotated as transit amplifying cells. Each of these superclusters was split by genotype and merged into in silico bulk experiments.

### Organoid culture

Crypts were isolated from adult mouse small intestine (~3 mo old) according to Sato & Clevers (2013). The crypts were seeded in Matrigel (Corning) and cultured in ENR medium (Advanced DMEM/F12 (Thermo Fisher Scientific), 1x penicillin–streptomycin (Thermo Fisher Scientific), 1x GlutaMAX (Thermo Fisher Scientific), 10 mM Hepes (Thermo Fisher Scientific), 1x B27 and 1x N2 (Thermo Fisher Scientific), 1.25 mM N-acetylcysteine (Sigma-Aldrich), 50 ng/ml of murine recombinant epidermal growth factor (R&D), 100 ng/ml recombinant murine Noggin (R&D), and 1 mg/ml recombinant murine R-spondin (R&D)). For establishing organoids from the E16.5 embryonal intestine, mating was set up and the day of the mating plug was taken as E0.5. The pregnant females were euthanized to get E16.5 embryos. For one experiment, 3–10 embryos of each genotype were used. The whole small intestine of the embryo was removed and cut into small pieces under a dissecting stereomicroscope (Leica). The pieces were rinsed with cold PBS and incubated with 20 ml of cold 2 mM EDTA at 4°C for 20 min. The epithelium was isolated by shaking, and fractions were collected as described for crypt isolation from adult mice. The epithelial fragments were seeded in Matrigel and cultured in ENR medium. MYC inhibitor 10058-F4 was purchased from MedChemExpress (HY-12702).

### Re-epithelialization of intestinal scaffolds with crypts

The ileum from the small intestine of mice was decellularized as previously described by Iqbal et al (2025). Pieces of decellularized intestines (dECMs) were primed at 37°C with 50 μl of media containing Advanced DMEM/F12, 1x penicillin–streptomycin, 1x GlutaMAX

(Thermo Fisher Scientific), 10 mM Hepes (Thermo Fisher Scientific), 1x B27 (Life Technologies), 1x N2 (Life Technologies), 1 mM N-acetylcysteine (Sigma-Aldrich), 50 ng/ml of murine recombinant epidermal growth factor (R&D), 100 ng/ml recombinant murine Noggin (PeproTech), 500 ng/ml recombinant murine R-spondin (R&D), and 10 $\mu$M Y-27632 (Sigma-Aldrich), for 15–60 min. This was followed by aspirating media carefully and plating around 100 crypts on each scaffold piece and incubating at 37°C for 10 min. Next, 10 μl of media was added to each scaffold piece and incubated at 37°C for 10 min, followed by an additional 20 μl per scaffold piece and incubation at 37°C for 20 min. Finally, 260 μl of media was added to each scaffold piece and put in the incubator at 37°C. The day after seeding, media were changed, and each scaffold piece was lifted to make it float. The second day after seeding, the media were changed to media without Y-27632. Media were subsequently replaced every other day. Re-epithelialized dECMs were fixed with 4% PFA at day 16 after plating. The re-epithelialized dECMs were analyzed using immunofluorescence. Primary antibody pan-laminin (1:300, 11575; Abcam) and secondary antibody Alexa 647–conjugated anti-rabbit (1:1,000; Life Technologies) were used. For nucleus detection, DAPI (1:1,000; Life Technologies) was used. Images were obtained with a Zeiss LSM980-Airy2 microscope using a 20x objective, NA = 08.

## Data Availability

Sequencing data are available from GEO accession number GSE244298.

## Supplementary Information

## Acknowledgements

The authors wish to thank Prof. Björn Rozell for his expert inputs on the study, Drs Rong Yu, Yujiao Wu, and Ekaterina Morgunova for their comments on the article, and Dr. Kashyap Dave for help with mouse crosses. We also wish to thank the core facilities at KI: The Unit for Morphological Phenotype Analysis (FENO), Bioinformatics and Expression Analysis Core Facility (BEA), Biomedicum Imaging Core (BIC), and Biomedicum Flow Cytometry (BFC). The study was supported by grants from the Swedish Research Council (D0815201) and Cancerfonden (20 1116 PjF 02H, 232693 Pj 01H).

### Author Contributions

I Sur: conceptualization, supervision, investigation, and writing—original draft, review, and editing.
W Zhao: investigation and writing—review and editing.
J Zhang: supervision, investigation, and writing—review and editing.
M Kling Pilström: investigation and writing—review and editing.
AT Webb: investigation and writing—review and editing.
H Cheng: investigation and writing—review and editing.

A Ristimäki: supervision, investigation, and writing—review and editing.

P katajisto: supervision, investigation, and writing—review and editing.

M Enge: supervision, investigation, and writing—review and editing.

H Rannikmae: investigation, and writing—review and editing.

M de la Roche: conceptualization, supervision, funding acquisition, investigation, and writing—original draft, review, and editing.

J Taipale: conceptualization, supervision, funding acquisition, investigation, and writing—original draft, review, and editing.

## Conflict of Interest Statement

I Sur and J Taipale are in the process of filing a patent application related to this work. Other authors declare no competing interests.

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
