## [Reviewer comments · Life Science Alliance]

Life Science Alliance

Shared requirement for MYC upstream super-enhancer region in tissue regeneration and cancer

Inderpreet Sur, Wenshuo Zhao, Jilin Zhang, Margareta Kling Pilström, Anna Webb, Huaitao Cheng, Ari Ristimäki, Pekka katajisto, Martin Enge, Helena Rannikmae, Marc de la Roche, and Jussi Taipale

DOI: <https://doi.org/10.26508/lsa.202403090>

Corresponding author(s): Jussi Taipale, Karolinska Institutet

Review Timeline:	Submission Date:	2024-10-14
	Editorial Decision:	2024-10-15
	Revision Received:	2025-02-28
	Editorial Decision:	2025-03-03
	Revision Received:	2025-03-14
	Accepted:	2025-03-14

Transaction Report:

Please note that the manuscript was previously reviewed at another journal and the reports were taken into account in the decision -making process at *Life Science Alliance*.

Referee #1 Review

Report for Author:

In this manuscript by Dr Taipale and collaborators, the Authors set out to address the role of the "8q24 Myc super-enhancer" in the intestine by taking advantage of a knock-out mouse model, Myc Δ 2-540/ Δ 2-540. The Authors show that while the mutant mice are normal, they have a compromised response to tissue regeneration and, to some extent, are also tumor-protected. The Authors propose that the 8q24 Myc super-enhancer is (i) the link between regenerative programs and tumorigenesis and (ii) a master regulator of tissue regeneration that explains the difference between tissue self-renewal and regeneration.

The manuscript reports solid phenotypic evidence of the role of the Myc-super-enhancer in regeneration and cancer (notwithstanding some aspects that need to be further detailed; see points below) but fails to provide convincing mechanistic insight. It remains unclear what the role of Myc in intestinal stem cells is, what are the key transcriptional programs controlled by Myc in these cells, and whether tumor protection and failure to regenerate are due to defects in the same population of cells. Thus, the manuscript does not explain how Myc regulation by its distal enhancer accounts for defects in regeneration and tumor protection.

As a general comment, many of the concepts presented in this manuscript have already been put forward: there is considerable evidence supporting similarities of regenerative-fetal programs with oncogenic programs, as well as data concerning the existence of a population of regenerative stem cells hijacked during tumor development. Thus, I would suggest the Authors to rephrase some of the conclusions and cite the relevant prior literature. As an example, the sentence "Our results establish Myc2-540 as the genetic link between tissue regeneration and tumorigenesis, and establish that normal tissue renewal and regeneration of tissues after severe damage are mechanistically distinct" (lines 33-35), should be re-edited to take into account previous works.

Major points

1. It is unclear to what extent mutant mice have defective regeneration in the colon. The data shown (fig 1B, i.e. weight loss) is suggestive but is an indirect assessment, while the IHC pictures in fig 1C are indicative but not quantitative or supported by statistics. Can the Authors assess the regeneration of the intestine in a direct and quantitative way? Is regeneration just delayed or is it impaired? Are mutant mice radiosensitive?

2. It is unclear to what extent residual levels of Myc are indeed contributing to gene transcription and whether Myc target genes expression is completely blunted or altered for all targets. Are all Myc-activated genes strongly/significantly downregulated in Myc Δ 2-540/ Δ 2-540 cells? What happens to Myc repressed genes? Also, if only some of the Myc targets are affected, is there an explanation for why only some of the targets are affected by Myc loss? Are these lowly expressed genes or low-affinity Myc targets? Please refer to published work to gain insight into this.

3. Fig2A reports expression changes for a set of Myc target genes in support of the claim that Myc Δ 2-540/ Δ 2-540 cells do not activate Myc targets. Yet the figure shows that some of the targets are UPREGULATED in Myc Δ 2-540/ Δ 2-540. Related to this, I would suggest also specifying whether the Myc target genes considered here are all genes activated by Myc. In case the Authors are also reporting Myc repressed genes, I would suggest reporting the activated and repressed genes separately. To this end, it would be relevant to assess what happens to Myc repressed genes.

4. There is ample evidence that in the intestine, there are at least two populations of stem cells that are engaged during homeostasis (Igr5+) and regeneration (revival stem cells). What is the role of Myc here? Can the Authors identify revival stem cells in their mutant mice? Are these cells different from those identified in WT animals? Is

the super-enhancer involved in the generation of the revival SCs or their conversion to the LGR5+?

By taking advantage of their scRNA-sq datasets, the Authors need to assess the presence of revival stem cells (based on published signatures) and their proficiency in key signaling pathways (YAP, NFkB, TGFbeta, IFN-gamma).

Also, the Lgr5+ stem cells should be characterized in more detail. They seem to increase in mutant mice (fig3E, clusters SSC3,4,5), can the Author infer, based on scRNA seq, why they increase? Also, since in this figure data are reported as %, it would be advisable to also assess absolute numbers in order to verify if this increment is absolute or fractional.

5. In fig 3E the Authors show subsets of intestinal stem cells and the expression of "common marker genes".

Please: (1) report expression values of common markers for wt and mutant cells (it is unclear what values are reported in the heatmap). (2) for each subset of stem cells report the differentially expressed genes (wt vs mutant-cells). (3) generate a single UMAP reporting wt and mutant cells, and highlight wt and mutant stem cell subsets in order to show whether or not mutant stem cells are indeed superimposed to wt cells. (4) among the common marker genes there are genes associated to cell proliferation: *ccna2*, *mki67*, *cdk1*, *top2a*. Are these differentially expressed in wt vs mutant cells? Given the loss of proliferation of mutant cells (post-irradiation), a change in their expression would be expected.

6. Pseudo-timing or similar analytical strategies should be used to assess stem cells' fate and differentiation trajectories during regeneration in mutant and wt-mice. This may reveal what population of cells/regenerative process is defective in mutant mice.

7. The Authors claim that *Myc* Δ 2-540/ Δ 2-540 does not alter cells in homeostatic conditions. Among other data, in support of this, they show that scRNA seq analysis identifies similar cell types on both wt and mutant cells (FigS2A). This should be complemented by differential expression analysis to evaluate whether these cells express genes differentially expressed in *Myc* Δ 2-540/ Δ 2-540 cells compared to wt cells.

8. Lines 258-263. The text reads: "Many of these cells appeared normal in morphology, suggesting that the tumor is capable of damaging surrounding normal tissue and inducing a wound-healing response. Normal-looking crypts expressing Ly6a (*Sca1*) at the tumor boundary did not express detectable transcripts of *Myc*, intestinal stem cell marker *Lgr5* or *Notum*, which is a specific marker of *Apc*-null cells (Fig. 4E), suggesting that these cells are at a growth disadvantage relative to the *Myc*-expressing tumor cells (Di Giacomo et al, 2017; Johnston, 2014)." From this, it seems that the tumors in *Apc*^{Min/+}; *Myc* Δ 2-540/ Δ 2-540 mice are expressing more *Myc* than the surrounding colonic cells. Is this correct? What is the level of *Myc*? How is *Myc* upregulated in these tumors (considering that one major claim of this manuscript is that mutant cells do not upregulate *Myc* during either regeneration or tumorigenesis)?

9. Figure S4: please provide a quantitative assessment of the budding phenotype and a statistical analysis.

Minor points:

1. Is Ly6a a *Myc* target gene?

2. Line 231-233: do the Authors have a post-mortem pathology report for these mice? What are the "symptoms consistent with intestinal tumorigenesis mentioned? What is the penetrance of late colonic tumors in *Apc*^{Min/+}; *Myc* Δ 2-540/ Δ 2-540 mice?

3. Paragraph 2, please use a more appropriate title since no causal relationship is reported here.

Referee #2 Review

Report for Author:

The manuscript investigates the role of the 8q24 super-enhancer region, located upstream of the Myc gene, in regulating intestinal stem cell mobilization during tissue regeneration and tumorigenesis. The authors find that while this enhancer is not essential for normal homeostasis, it is critical during regenerative responses to radiation damage and for the growth of adult intestinal cells in vitro. Deletion of the enhancer significantly impairs regeneration, leading to ineffective recruitment of regenerative stem cells and failure to restore intestinal crypt architecture. Additionally, the study identifies Myc2-540 as a genetic link between regeneration and tumorigenesis, noting that its deletion extends the lifespan of ApcMin/+ mice without affecting tumor gene expression profiles.

Overall, while the manuscript presents interesting findings regarding the 8q24 super-enhancer's role in tissue regeneration and tumorigenesis, it has significant shortcomings. The established importance of Myc in these processes, alongside recent publications detailing its role in Wnt oncogenic transcription (PMID: 38971196), diminishes the overall novelty of the work. The lack of discussion on key signaling pathways, inadequate statistical analysis, and unclear figure presentations detract from the study's impact.

Major concerns:

1. **Lack of Contextual Discussion:** The manuscript does not address relevant studies on the Wnt-Myc (PMID: 20708588) and PAF-Myc (PMID: 29533773) signaling pathways, both of which are critical for intestinal regeneration and tumorigenesis. The essential role of Myc overexpression in these processes is also overlooked (PMID: 38971196). As the function of Myc expression per se has been well studied, and that the role of super-enhancers on regulating gene expression is known, the contribution of the study mainly lies in demonstrating the effect in vivo in the setting of cancer.
2. **Insufficient Quantification of Histological Data:** The analysis of histological phenotypes lacks adequate quantification, and statistical significance is not clearly indicated. Numerous figure panels contain only photomicrographs of stains without quantification. Moreover, in readouts such as Ki67 IHC, the crypts selected as representative stains do not appear to be well oriented. As such, it is unclear what the effects are. The authors tend to use RNA expression, obtained by sequencing, as the "definitive" quantification, and "validated" using protein expression readouts such as IHC. While this is acceptable, the IHC still requires quantification.
3. **Poorly Described Figures:** Figures 4C-E are inadequately described, with unclear data presentation. For example, where is the tumor front? Are these adenomas only or invasive carcinomas? It is unclear whether a pathologist who is experienced in this type of analysis was involved in the study, but it will definitely improve quantification and interpretation.
4. **scRNA-seq Analysis:** The clustering for scRNA-seq analysis (Figures 3D and 3G) is not optimally justified and unconvincing compared to the cited reference (PMID: 34079126).

Minor Suggestions:

1. Include sample data points in quantification graphs for enhanced clarity.
2. Add graphical indicators on histology images for better interpretation.
3. Reduce excessive speculation and inference in the results sections.
4. Revise the conclusion figure (Figure 5) to enhance clarity and avoid confusion.

Referee #3 Review

Report for Author:

Cancer has often been described as a wound that never heals. While malignant cells differ morphologically from normal proliferating cells, they share many characteristics with tissues that are healing or regenerating. The reasons behind this similarity have been unclear. This study reveals that the genomic region upstream of the Myc gene, which is not essential for normal tissue maintenance but is crucial for intestinal tumor development, is also necessary for intestinal regeneration following radiation damage. The paper presents interesting findings

but lacks depth in explaining key results. Important concepts, such as the relationship between MYC enhancer depletion and regeneration, need more emphasis based on existing literature.

Detailed comments by Figure:

Figure 1:

1D: Include daily images and quantifications from 0d-5d for clarity, especially for Ki67 levels between 3d-5d in WT, which may correlate with MYC changes.

1D: Consider examining proliferation rates from 5d-7d post-IR to see if they correlate with MYC decrease.

1E: Provide additional images for 5 days of MYC depletion; the differences at 0d-3d are not visually distinct.

Figure 2:

PVT1 (106-107): The conclusion regarding PVT1's role is weak; additional experiments, such as PVT1 depletion during IR, could strengthen the argument. Recent literature suggests a more complex relationship with MYC.

2D: Include levels of 4EBP1 before and after IR for completeness.

2E: Quantification of phosphorylated proteins should accompany the scRNA-seq data for clarity.

Show mTOR levels to provide further context.

Figure 3:

Sup Fig. 3C: Important data that should be included in Fig. 3B for a comprehensive view.

Sup Fig. 3D: Clarify that the late organoids are those discussed in Sup Fig. 3C after passage 18.

S4B: Include images for all conditions similar to A.

181-183: The comparable morphology of intestines in different conditions raises concerns that should be addressed.

205-207: Clarify how the conclusions on proliferation, differentiation, and crypt-like structures were drawn from the data.

3G: the definition of specific clusters is unclear and the strategy for identification needs to be clarified.

Particularly since the distinction of those clusters is not simply visible in the presented UMAPs.

Figure 4:

4D & Sup Fig. 5B: Include control images for all comparisons.

Leverage scRNA-seq data to support transcriptional changes in Fig. 4E and address previously discussed stem cell populations.

4E: Question the choice of ISH over IHC; clarify the rationale behind focusing on specific changes, like Ly6a, which is critical yet not well addressed. Representative areas needs quantification, particularly since the bottom panel shows a rather cross sectioning area of the tumors.

4F & 4G: The purpose of the scRNA-seq controls is unclear; more relevant data could enhance understanding of the findings.

Figure 5

Should be checked for typos and clarity of the findings needs to be enhance. It contains various aspects that have not been demonstrated in this paper like increase in mass and somatic stem cells. Again clarity with regards to the actual finding is difficult to define from the this figure.

October 15, 2024

Re: Life Science Alliance manuscript #LSA-2024-03090-T

Prof. Jussi Taipale
UNIVERSITY OF HELSINKI
GENOME-SCALE BIOLOGY RESEARCH PROGRAM
P.O. BOX 63 (HAARTMANINKATU 8)
Novum, Elevator F, 5th floor
HELSINKI FI00014 UNIVERSITY OF HELSINKI
Finland

Dear Dr. Taipale,

Thank you for submitting your manuscript entitled "The 8q24 super-enhancer region is required for MYC-dependent mobilization of regenerative stem cells" to Life Science Alliance. We invite you to submit a revised manuscript addressing the following Reviewer comments:

- Address Reviewer 1's major points #1, 2, 3, 5, 7 & 9, as well as the minor points. Major point #6 would be interesting, but is optional to address.
- Address Reviewer 2's comments.
- Address Reviewer 3's comments.

Thank you for this interesting contribution to Life Science Alliance. We are looking forward to receiving your revised manuscript.

Sincerely,

B. MANUSCRIPT ORGANIZATION AND FORMATTING:

Manuscript: LSA-2024-03090-TR

Below is our point-by-point response to all specific criticisms of each reviewer. The reviewers comments are in *italic* and our response in regular Roman. Specific changes made to the manuscript are in Roman **bold**.

Referee #1:

In this manuscript by Dr Taipale and collaborators, the Authors set out to address the role of the "8q24 Myc super-enhancer" in the intestine by taking advantage of a knock-out mouse model, Myc Δ 2-540/ Δ 2-540. The Authors show that while the mutant mice are normal, they have a compromised response to tissue regeneration and, to some extent, are also tumor-protected. The Authors propose that the 8q24 Myc super-enhancer is (i) the link between regenerative programs and tumorigenesis and (ii) a master regulator of tissue regeneration that explains the difference between tissue self-renewal and regeneration.

The manuscript reports solid phenotypic evidence of the role of the Myc-super-enhancer in regeneration and cancer (notwithstanding some aspects that need to be further detailed; see points below) but fails to provide convincing mechanistic insight. It remains unclear what the role of Myc in intestinal stem cells is, what are the key transcriptional programs controlled by Myc in these cells, and whether tumor protection and failure to regenerate are due to defects in the same population of cells. Thus, the manuscript does not explain how Myc regulation by its distal enhancer accounts for defects in regeneration and tumor protection.

As a general comment, many of the concepts presented in this manuscript have already been put forward: there is considerable evidence supporting similarities of regenerative-fetal programs with oncogenic programs, as well as data concerning the existence of a population of regenerative stem cells hijacked during tumor development. Thus, I would suggest the Authors to rephrase some of the conclusions and cite the relevant prior literature. As an example, the sentence "Our results establish Myc²⁻⁵⁴⁰ as the genetic link between tissue regeneration and tumorigenesis, and establish that normal tissue renewal and regeneration of tissues after severe damage are mechanistically distinct" (lines 33-35), should be re-edited to take into account previous works.

We thank the reviewer for the comment and **have now rephrased the conclusions and cited relevant prior literature (also pointed out by reviewer #2) on page15, lines 326-327. The sentences that were on the lines 33-35 of the abstract now read:** " Our results establish a function for the Myc²⁻⁵⁴⁰ super-enhancer region as the

genetic link between tissue regeneration and tumorigenesis, and establish that normal tissue renewal and regeneration of tissues after severe damage are mechanistically distinct”.

Major points

1. It is unclear to what extent mutant mice have defective regeneration in the colon. The data shown (fig 1B, i.e. weight loss) is suggestive but is an indirect assessment, while the IHC pictures in fig 1C are indicative but not quantitative or supported by statistics. Can the Authors assess the regeneration of the intestine in a direct and quantitative way? Is regeneration just delayed or is it impaired? Are mutant mice radiosensitive?

We thank the reviewer for this comment. We have now quantified the damage and repair in the intestine with respect to crypt depth and villus height. These parameters are widely used in the field to measure intestinal damage and repair. **The quantification data is now presented in panel D of the revised Fig 1. It shows a significantly reduced regeneration capacity of $Myc^{\Delta 2-540/\Delta 2-540}$ intestines, as evidenced by decreased crypt depth ($p < 0.0001$) and villus height ($p < 0.0001$) compared to wild-type.**

$Myc^{\Delta 2-540/\Delta 2-540}$ mice irradiated with 12 Gy do not show signs of recovery within the 5d span of the experiment. At the end of 5d $Myc^{\Delta 2-540/\Delta 2-540}$ mice need to be euthanised for ethical reasons. The mice are therefore highly radiosensitive. **We have now clarified this in the manuscript (Page 4, lines 75-80).**

2. It is unclear to what extent residual levels of Myc are indeed contributing to gene transcription and whether Myc target genes expression is completely blunted or altered for all targets.

Regarding residual levels of MYC, $Myc^{\Delta 2-540/\Delta 2-540}$ intestine has only 1.4% (transit amplifying cells)-2.8% (stem cells) of normal *Myc* levels which alters MYC target gene expression but does not completely blunt it. Whether this expression is due to additional transcription factors controlling MYC target gene expression or due to residual MYC levels cannot be ascertained from the present data and would require additional experiments which are beyond the scope of this study.

Are all Myc-activated genes strongly/significantly downregulated in $Myc^{\Delta 2-540/\Delta 2-540}$ cells? What happens to Myc repressed genes?

Majority, but not all, of the 126 functionally conserved MYC target genes (Zeilke et al. 2022 PMID: 35472319) are downregulated in $Myc^{\Delta 2-540/\Delta 2-540}$ cells after irradiation (89% of the genes are downregulated in stem cells (SC) and 84% in transit amplifying cells (TA). The effect is significant ($p < 0.05$) for 64% and 62% of the genes, respectively. This is consistent with most (~90%) of the conserved MYC targets being induced by MYC (Zielke et al 2022 <https://doi.org/10.1016/j.devcel.2022.03.018>). **We have now provided this information in the manuscript (Page 6; Lines 117-121).**

The contribution of MYC repressed targets is also an interesting question for which we thank the reviewer. We have now analyzed expression of validated c-MYC repression targets (Diamant et al 2025 <https://doi.org/10.1093/nar/gkae1080>).

https://maayanlab.cloud/Harmonizome/gene_set/Validated+targets+of+C-MYC+transcriptional+repression/PID+Pathways

Consistently with these targets being MYC repressed, SC and TA cells from wild-type mice express less of this set compared to the same cells from $Myc^{\Delta 2-540/\Delta 2-540}$ mice. However, irradiation does not induce further repression of this set. This is in contrast to the conserved MYC- targets whose expression after irradiation is dramatically increased in the wildtype compared to the $Myc^{\Delta 2-540/\Delta 2-540}$ mice. **We have now also included MYC target expression data for targets repressed by MYC (Panels C in new Fig 3). We have also clarified this in the manuscript (Page6, lines 124-126).**

Also, if only some of the Myc targets are affected, is there an explanation for why only some of the targets are affected by Myc loss? Are these lowly expressed genes or low-affinity Myc targets? Please refer to published work to gain insight into this.

The extent of $MYC^{\Delta 2-540}$ effect varies amongst the targets. This is not dependent on the level of expression (now shown in Supplementary Table S3). We also looked at the presence of high affinity canonical E-box sequence in the promoters of the conserved MYC targets. Of the 55 genes with a positive log₂ FC of >1 in stem cells of WT_{IR} vs $Myc^{\Delta 2-540/\Delta 2-540}$ _{IR} after irradiation (2dpi), 75% have a canonical E-box. This percentage drops to ~50% when we look at the conserved target genes whose expression is decreased in the WT_{IR} stem cells compared to $Myc^{\Delta 2-540/\Delta 2-540}$ _{IR}. Thus although there is a trend towards presence of high affinity MYC binding sites and increased expression, we feel that a conclusive answer to whether affinity of binding sites determines the response of the target gene to MYC perturbation in our experiment would require extensive computational and MYC binding analysis which we feel is beyond the scope of the present work. Alternatively, it is possible that tissue-specific mechanisms determine the response of MYC targets. To avoid undue speculation we prefer not to discuss in the manuscript the reason why some targets are affected more

than others due to MYC loss. **Instead, we have included the expression data itself (Supplementary Table S3) to facilitate future work.**

3. Fig2A reports expression changes for a set of Myc target genes in support of the claim that Myc Δ 2-540/ Δ 2-540 cells do not activate Myc targets. Yet the figure shows that some of the targets are UPREGULATED in Myc Δ 2-540/ Δ 2-540. Related to this, I would suggest also specifying whether the Myc target genes considered here are all genes activated by Myc. In case the Authors are also reporting Myc repressed genes, I would suggest reporting the activated and repressed genes separately. To this end, it would be relevant to assess what happens to Myc repressed genes.

We thank the reviewer for this comment. We think the reviewer refers to old Fig 2B. Upon reanalysis, we realised that there was an error in labelling of this figure panel, the y-axis was mislabelled. **We have now corrected the label of the y-axis of this panel (new Fig 3B) to indicate that what is shown is WT vs Myc Δ 2-540/ Δ 2-540 log fold change for both irradiated and non-irradiated samples. We additionally separately provide data on MYC-repressed targets (new Fig 3C and supplementary Table S4).** We thank the reviewer for diligence and pointing this out.

4. There is ample evidence that in the intestine, there are at least two populations of stem cells that are engaged during homeostasis (lgr5+) and regeneration (revival stem cells). What is the role of Myc here? Can the Authors identify revival stem cells in their mutant mice? Are these cells different from those identified in WT animals? Is the super-enhancer involved in the generation of the revival SCs or their conversion to the LGR5+?

By taking advantage of their scRNA-sq datasets, the Authors need to assess the presence of revival stem cells (based on published signatures) and their proficiency in key signaling pathways (YAP, NFkB, TGFbeta, IFN-gamma).

Also, the lgr5+ stem cells should be characterized in more detail. They seem to increase in mutant mice (fig3E, clusters SSC3,4,5), can the Author infer, based on scRNA seq, why they increase? Also, since in this figure data are reported as %, it would be advisable to also assess absolute numbers in order to verify if this increment is absolute or fractional.

This is a very interesting and exciting line of enquiry suggested by the reviewer. However, we feel that detailed analysis of stem cell types is beyond the scope of the present study.

5. In fig 3E the Authors show subsets of intestinal stem cells and the expression of "common marker genes". Please:

(1) report expression values of common markers for wt and mutant cells (it is unclear what values are reported in the heatmap).

We thank the reviewer for the suggestion and **have provided the requested data as Supplementary Table S8 and referred to it in the text (Page10, lines 230-231).**

(2) for each subset of stem cells report the differentially expressed genes (wt vs mutant-cells).

We thank the reviewer for the comment and **have provided results of differential expression analysis (wild-type vs $Myc^{\Delta 2-540/\Delta 2-540}$) for each subset of fetal stem cells from organoid cultures as Supplementary Table S7 and referred to it in the text (Page10, lines 230-231).**

(3) generate a single UMAP reporting wt and mutant cells, and highlight wt and mutant stem cells subsets in order to show whether or not mutant stem cells are indeed superimposed to wt cells.

We thank the reviewer for this comment. We have now generated a single UMAP reporting both the WT and mutant cells. We also calculated the mean value of x and y of all the cells belonging to that subcluster to find the center and marked it with a circle on the UMAP (fixed radius of 1.5) showing that cells from the two genotypes predominantly superimpose. **The revised UMAPs are presented in new Fig 5. We have also added information on annotations on the UMAP to the figure legend (new Fig 5) as well as in materials and methods on Page20, lines 459-462.**

(4) among the common marker genes there are genes associated to cell proliferation: *ccna2*, *mki67*, *cdk1*, *top2a*. Are these differentially expressed in wt vs mutant cells? Given the loss of proliferation of mutant cells (post-irradiation), a change in their expression would be expected.

We agree that this is an important question. The fetal 3D-cultures from $Myc^{\Delta 2-540/\Delta 2-540}$ intestine prominently differ from the adult organoids (also damage repair and cancer) in that they are able to proliferate and increase in size as spheroids. $Myc^{\Delta 2-540/\Delta 2-540}$ cells belonging to stem cell subclusters that are enriched in WT budding organoids (SSC1_F and SSC3_F) do show mostly lower expression of *ccna2*, *mki67*, *cdk1*, *top2a* but only lower expression of *Ccna2* (logFC=-0,293788, $p = 0.03$) and *Top2a* (logFC=-0,8393516, $p=0,0013754$) in SSC3_F is statistically significant. **We have provided the differential expression of the proliferation genes (Ccna2, Mki67, Cdk1, Top2a) as part of**

Supplementary Table S8. We have also discussed it in the manuscript (Page 10, lines 225-227).

6. Pseudo-timing or similar analytical strategies should be used to assess stem cells' fate and differentiation trajectories during regeneration in mutant and wt-mice. This may reveal what population of cells/regenerative process is defective in mutant mice.

We agree with the reviewer that deciphering how the stem cell fates and differentiation trajectories are affected is important. We thank the reviewer for the comment but feel that detailed analysis of the stem cells is beyond the scope of the current work.

7. The Authors claim that $Myc^{\Delta 2-540/\Delta 2-540}$ does not alter cells in homeostatic conditions. Among other data, in support of this, they show that scRNA seq analysis identifies similar cell types on both wt and mutant cells (FigS2A). This should be complemented by differential expression analysis to evaluate whether these cells express genes differentially expressed in $Myc^{\Delta 2-540/\Delta 2-540}$ cells compared to wt cells.

We thank the reviewer for the comment and have now provided the differential expression analysis (WT vs $Myc^{\Delta 2-540/\Delta 2-540}$) for each of the cell types. With a threshold of $\log_{2}FC > 1$ or $\log_{2}FC < -1$, and $p < 0.05$, we find following number of genes whose expression is altered e.g. in the Stem cells=457 genes, Transit amplifying cells=464 genes and Enterocyte=243 due to the deletion. Based on analysis using MA plots (new Supplementary Fig S3) the top genes in the differentially expressed list, ranked based on expression (Average $\log_{2}[\text{Mean (Normalized readcounts)} + 1]$), were genes involved in ribosome biogenesis in SC and TA populations, consistent with regulation of this target gene set by MYC.

Despite the transcriptional changes, intestinal proliferation and differentiation during homeostasis is not affected by loss of Myc^{2-540} . This is in line with earlier reports in the field, including our data, that MYC is dispensable for adult intestinal homeostasis (Bettess et al (2005) PMID: 16107730; Dave et al 2017: PMID: 28583252). **We have now included the results of differential expression analysis for each cell type as a new Supplementary Table S6 and the MA plots are provided as new Supplementary Fig S3. The results of differential expression analysis are also discussed in the text (Page7, lines 152-156).**

8. Lines 258-263. The text reads: "Many of these cells appeared normal in morphology, suggesting that the tumor is capable of damaging surrounding normal tissue and inducing a wound-healing response. Normal-looking crypts expressing Ly6a (Sca1) at the tumor boundary did not express detectable transcripts of Myc, intestinal

stem cell marker *Lgr5* or *Notum*, which is a specific marker of *Apc*-null cells (Fig. 4E), suggesting that these cells are at a growth disadvantage relative to the *Myc*-expressing tumor cells (Di Giacomo et al, 2017; Johnston, 2014)." From this, it seems that the tumors in *Apc*^{Min/+}; *Myc*^{Δ2-540/Δ2-540} mice are expressing more *Myc* than the surrounding colonic cells. Is this correct? What is the level of *Myc*? How is *Myc* upregulated in these tumors (considering that one major claim of this manuscript is that mut-cells do not upregulate *Myc* during either regeneration or tumorigenesis?)

This is an interesting question and we thank the reviewer for it. It is correct that the normal epithelium of *Myc*^{Δ2-540/Δ2-540} and *Apc*^{Min/+}; *Myc*^{Δ2-540/Δ2-540} intestine expresses only very low levels of MYC. We also do not see induction of expression of *Myc* during regeneration. It is also correct that tumors in old *Apc*^{Min/+}; *Myc*^{Δ2-540/Δ2-540} mice do express *Myc*. We can only conjecture at present that additional regulatory elements (e.g. those located on the 3' end of MYC) regulate expression of *Myc* in tumors that develop in *Apc*^{Min/+}; *Myc*^{Δ2-540/Δ2-540} mice. To conclusively show the mechanism of how *Myc* is activated in the tumors will require extensive genome-sequencing and mutational analysis etc. which is beyond the scope of this study. **We have clarified this in the manuscript (Page13, lines 299-301).**

9. Figure S4: please provide a quantitative assessment of the budding phenotype and a statistical analysis.

We thank the reviewer for the comment. Quantification of the phenotype shows that WT cultures treated with DMSO consist of ~90% budding organoids and ~10% spheroids, cultures treated with *Myc*-inhibitor 10058-F4 (100 μM) had on an average ~30% budding organoids and ~70% spheroids. **We have revised this figure (Supplementary Fig S4) and included also the quantification of the phenotype.**

Minor points:

1. Is *Ly6a* a *Myc* target gene?

We thank the reviewer for this question. Although *Ly6a* (*Sca1*) is present on the same chromosome as *Myc*, to our knowledge, it has not been reported to be a MYC target in literature. Additionally, we do see expression of *Ly6a* (*Sca1*) for instance in normal looking crypts next to the tumor boundary (Figure 6E) in cells that do not express detectable *Myc* by ISH, suggesting that *Ly6a* expression is independent of MYC. **We have now clarified this in the manuscript (Page13, lines 295-299).**

2. Line 231-233: do the Authors have a post-mortem pathology report for these mice?

What are the "symptoms consistent with intestinal tumorigenesis mentioned? What is the penetrance of late colonic tumors in ApcMin/+; Myc Δ 2-540/ Δ 2-540 mice?

We regret that we do not have post-mortem pathology report for all these mice. The symptoms consistent with intestinal tumorigenesis refer to rectal prolapse, bleeding, weight loss and/or abdominal bloating. **We have clarified this in the manuscript (Page11, lines 255-258)**. Although analysis of all the mice in the study is still ongoing, our estimate is that approximately 46% of mice (500 days or older) develop these tumors. Due to the incompleteness of the data we have not included it in the manuscript.

3.Paragraph 2, please use a more appropriate title since no causal relationship is reported here.

We thank the reviewer for the comment and **have changed the subtitle of the second paragraph to "Failure to regenerate is associated with loss of induction of MYC" (Page5, line 105)**.

Referee #2:

The manuscript investigates the role of the 8q24 super-enhancer region, located upstream of the Myc gene, in regulating intestinal stem cell mobilization during tissue regeneration and tumorigenesis. The authors find that while this enhancer is not essential for normal homeostasis, it is critical during regenerative responses to radiation damage and for the growth of adult intestinal cells in vitro. Deletion of the enhancer significantly impairs regeneration, leading to ineffective recruitment of regenerative stem cells and failure to restore intestinal crypt architecture. Additionally, the study identifies Myc2-540 as a genetic link between regeneration and tumorigenesis, noting that its deletion extends the lifespan of ApcMin/+ mice without affecting tumor gene expression profiles.

Overall, while the manuscript presents interesting findings regarding the 8q24 super-enhancer's role in tissue regeneration and tumorigenesis, it has significant shortcomings. The established importance of Myc in these processes, alongside recent publications detailing its role in Wnt oncogenic transcription (PMID: 38971196), diminishes the overall novelty of the work. The lack of discussion on key signaling pathways, inadequate statistical analysis, and unclear figure presentations detract from the study's impact.

Major concerns:

1. Lack of Contextual Discussion: The manuscript does not address relevant studies on the Wnt-Myc (PMID: 20708588) and PAF-Myc (PMID: 29533773) signaling pathways, both of which are critical for intestinal regeneration and tumorigenesis. The essential role of Myc overexpression in these processes is also overlooked (PMID: 38971196). As the function of Myc expression per se has been well studied, and that the role of super-enhancers on regulating gene expression is known, the contribution of the study mainly lies in demonstrating the effect in vivo in the setting of cancer.

We thank the reviewer for the above references and **we have now cited them in the manuscript (Page15, lines 326-328) along with additional literature on the role of Myc levels in these processes (<https://doi.org/10.1038/S41467-022-34079-x>)**. Our results are consistent with the data presented in the studies highlighted by the reviewer in which the role of MYC in driving regeneration and cancer has been documented along with demonstration of PAF being a regulator of MYC expression in these conditions. Our work expands these studies further by providing a function to a regulatory region of Myc that contains the highest cancer susceptibility burden in our entire genome. Even more importantly our result demonstrates a differential requirement for MYC in fetal vs adult models of regeneration. Understanding of the mechanisms governing this difference could provide ways of reverting cancer

phenotypes which in our view is a highly relevant and immediate need in the cancer field. **We have expanded our discussion to include the above points and cited the additional literature (Page15, lines 326-328).**

Regarding the reviewers comment on super-enhancers regulating gene expression: It is well known that super-enhancers regulate gene expression however their link to a function still needs to be established. **We have clarified in the text that although the super-enhancer region regulates MYC expression in specific tumors, its normal biological function has been unknown prior to this work (Page3, lines 48-49).**

2. Insufficient Quantification of Histological Data: The analysis of histological phenotypes lacks adequate quantification, and statistical significance is not clearly indicated. Numerous figure panels contain only photomicrographs of stains without quantification. Moreover, in readouts such as Ki67 IHC, the crypts selected as representative stains do not appear to be well oriented. As such, it is unclear what the effects are. The authors tend to use RNA expression, obtained by sequencing, as the "definitive" quantification, and "validated" using protein expression readouts such as IHC. While this is acceptable, the IHC still requires quantification.

We thank the reviewer for the comment. **We have revised the IHC figures to show well oriented crypts. We have also provided quantification as well as statistical analysis of IHC data pertaining to Ki-67 staining in Fig 1D (in new Fig 1F) and Supplementary Fig S2A (in revised Supplementary Fig S2B). We have also quantified the number of paneth, goblet and enteroendocrine cells from IHC data represented in Supplementary Fig S2A (in revised Supplementary Fig S2B).** For quantification of ISH data showing Lgr5 positive cells, we believe the best way would be to use Lgr5-egfp mice crossed to our strain where Lgr5 positive cells are marked by egfp. In the absence of such a cross and any Lgr5 specific antibody we leveraged our scRNA seq data to quantify the number of Lgr5 positive cells in wild-type and $Myc^{\Delta 2-540/\Delta 2-540}$ intestine. **We have added Lgr5 quantification data to the new Supplementary Fig S2B.**

3. Poorly Described Figures: Figures 4C-E are inadequately described, with unclear data presentation. For example, where is the tumor front? Are these adenomas only or invasive carcinomas? It is unclear whether pathologist who are experienced in this type of analysis was involved in the study, but it will definitely improve quantification and interpretation.

We thank the reviewer for the comment. **We have marked the tumor and normal tissue in the images.** The pathologist classified the tumors in 4 month old $Apc^{Min/+}$ mice as adenomas while the tumors shown from old $Apc^{Min/+}$; $Myc^{\Delta 2-540/\Delta 2-540}$ mice (new Fig

6) were classified as ileal adenocarcinoma that are invasive. The ISH images shown in the figure do not capture the tumor front (referring to the invasion of the tumor into the mucosa) but rather the sides of the tumor where the tumor is next to the normal epithelium. **To avoid confusion we have removed the term ‘invasive front’ and the sentence now reads:**

” *Apc*^{Min/+}; *Myc*^{Δ2-540/Δ2-540} tumors often have invaded radially to encircle the entire small intestine, and display a lateral ‘pushing edge’ growth pattern, instead of the finger-like pattern common to polyps” **(Page12, lines 269-271).**

4. *scRNA-seq Analysis: The clustering for scRNA-seq analysis (Figures 3D and 3G) is not optimally justified and unconvincing comparing to the cited reference (PMID: 34079126).*

We thank the reviewer for the comment. For Fig 3D our intention with subclustering was to determine if there was heterogeneity in the stem cell population that could distinguish the WT organoids from *Myc*^{Δ2-540/Δ2-540} 3D spheroids. The cited reference (Yum *et al* 2021) was used for the cell type specific marker gene to first annotate the known intestinal cell types. The stem cell population then underwent further hierarchical subclustering. In the new revised Fig 5A, we have now marked the location of the stem cell subclusters on the UMAP by calculating the mean value of x and y of all the cells belonging to that subcluster to find the center and then circling it on the UMAP (fixed radius of 1.5). For 3G we have revised the figure (new Fig 5D) and marked the clusters using the same strategy as for the subclusters. **We have modified the materials and methods section to include this information (Page20, lines 459-462).**

Minor Suggestions:

1. *Include sample data points in quantification graphs for enhanced clarity.*

We thank the reviewer for the comment. **We have provided data points in revised Fig 1D, 3F, Supplementary Fig S2 and S4) and error bars in barplots (new Fig 2B, Fig 4E, 6G and revised Supplementary Fig S1 and S2).**

2. *Add graphical indicators on histology images for better interpretation.*

We thank the reviewer for the comment. **We have provided better quality images together with graphical indicators in new Fig 1E, new Fig 2A, new Fig 6D-E).**

3. *Reduce excessive speculation and inference in the results sections.*

We thank the reviewer for the comment. We have now reduced excessive speculation and inference in the results section.

Specifically we have deleted the following

Page9, lines 203-207:suggesting that the growth of *Myc* ^{$\Delta 2-540/\Delta 2-540$} spheroids is supported by a stem cell pool that is different from that of the wildtype organoids. Furthermore, this stem cell pool can support proliferation and differentiation in the absence of MYC but is not proficient in developing crypt-like structures associated with adult intestine.

and rephrased the following

Page 10, lines 219-222:these results establish that although fetal proliferation and differentiation can occur in the absence of MYC, the adult repair and regenerative state is critically coupled to a MYC-dependent proliferation and mobilization of stem cells required for repair/regeneration.

To

....These results demonstrate that although fetal intestinal 3-D cultures can be propagated in almost complete absence of MYC, the adult repair and regenerative state is critically coupled to a MYC-dependent proliferation and mobilization of regenerative stem/progenitor cells (**Page11, lines 243-246**).

4. Revise the conclusion figure (Figure 5) to enhance clarity and avoid confusion.

We thank the reviewer for the comment and **have modified the conclusion figure** (new Fig 7). Model now describes how cancer susceptibilities-associated super-enhancer region *Myc* ^{$2-540$} drives regenerative growth in the fetal and adult intestine and in cancer. Since majority of the MYC targets are proteins involved in ribosomal biogenesis, the model further describes how different levels of protein synthesis can result in different proliferative cell fate choices ranging from normal homeostasis to regeneration and cancer.

Referee #3:

Cancer has often been described as a wound that never heals. While malignant cells differ morphologically from normal proliferating cells, they share many characteristics with tissues that are healing or regenerating. The reasons behind this similarity have been unclear. This study reveals that the genomic region upstream of the Myc gene, which is not essential for normal tissue maintenance but is crucial for intestinal tumor development, is also necessary for intestinal regeneration following radiation damage. The paper presents interesting findings but lacks depth in explaining key results. Important concepts, such as the relationship between MYC enhancer depletion and regeneration, need more emphasis based on existing literature.

Detailed comments by Figure:

Figure 1:

1D: Include daily images and quantifications from 0d-5d for clarity, especially for Ki67 levels between 3d-5d in WT, which may correlate with MYC changes.

We thank the reviewer for comments. We have IHC data for Ki-67 levels and ISH data for *Myc* expression for 3d and 5d after radiation (WT and mutant). We also have *Myc* expression data using qPCR for WT on d0, d1, d2, d3 and d5 of radiation. There was a steady increase in MYC expression from d0-d3 with peak expression occurring on d3. **We have provided this data in supplementary Fig S1. We have additionally included images of intestine showing Ki-67 staining 3d after irradiation (new Fig 1E) along with quantification of number of Ki-67 positive cells. We have also provided additional ISH images for *Myc* expression at d5 to better correlate the Ki-67 staining with *Myc* expression (new Fig 2).**

1D: Consider examining proliferation rates from 5d-7d post-IR to see if they correlate with MYC decrease.

We thank the reviewer for this comment. **We have now provided ISH images for *Myc* expression on d5** which are in agreement with our qPCR data showing that *Myc* expression decreases after d3 of radiation (new Fig 2). **We have also quantified the Ki-67⁺ cells on d5.** Unlike the decrease in *Myc* expression, the number of Ki-67⁺ cells on d5 remains higher than in the non-irradiated samples. **We have clarified this in the manuscript (Page4-5, lines 85-89).**

1E: Provide additional images for 5 days of MYC depletion; the differences at 0d-3d are not visually distinct.

We thank the reviewer for the suggestion. **We have provided *Myc* ISH images for d5 after radiation. We have also provided new images with longitudinally sectioned crypts to make visual inspection easier and clear (new Fig 2).**

Figure 2:

PVT1 (106-107): The conclusion regarding PVT1's role is weak; additional experiments, such as PVT1 depletion during IR, could strengthen the argument. Recent literature suggests a more complex relationship with MYC.

We appreciate the reviewers comments however we feel that investigation of PVT1 involvement merits another study.

2D: Include levels of 4EBP1 before and after IR for completeness.

We thank the reviewer for the comment. After irradiation, compared to wild-type, the expression of 4Ebp1 is significantly reduced in *Myc*^{Δ2-540/Δ2-540} in a pattern similar to that seen for phospho-4ebp1. **We have now provided the 4Ebp1 IHC data (new Fig 3) and modified the text (Page7, lines 142-146).**

2E: Quantification of phosphorylated proteins should accompany the scRNA-seq data for clarity.

We thank the reviewer for the comment. **We have quantified the IHC staining for phospho-4Ebp1 and this is provided in new Fig 3F.**

Show mTOR levels to provide further context.

We thank the reviewer for this comment. We have now examined the expression of mTOR in our scRNA-seq data after irradiation in *Myc*^{Δ2-540/Δ2-540} and wild-type cells. At 2 dpi, the wild-type cells express slightly higher levels of mTOR mRNA compared to *Myc*^{Δ2-540/Δ2-540} cells (log2 FC=0.246 in stem cells). Such a small change in mRNA is unlikely to explain changes in levels of phospho-Eif4ebp1 between these conditions. Instead, the change is consistent with MYC induction of Eif4ebp1 mRNA combined with TOR-dependent phosphorylation of the Eif4ebp1 protein (Hruby et al, 2024

doi: <https://doi.org/10.1101/2024.03.06.583558>; Tameire et al, 2019

DOI: [10.1038/s41556-019-0347-9](https://doi.org/10.1038/s41556-019-0347-9)). **We have clarified this in the manuscript (Pages7, lines 136-142).**

Figure 3:

Sup Fig. 3C: Important data that should be included in Fig. 3B for a comprehensive view.

We thank the reviewer for the insight. **We have moved the data presented in old Supplementary Fig 3C to the new main Fig 4D.**

Sup Fig. 3D: Clarify that the late organoids are those discussed in Sup Fig. 3C after passage 18.

We thank the reviewer for the comment. **We have clarified this in revised new Fig 4E.**

S4B: Include images for all conditions similar to A.

We thank the reviewer for the comment and **have provided a new revised Supplementary Fig S4 with images from the DMSO control.**

181-183: The comparable morphology of intestines in different conditions raises concerns that should be addressed.

We thank the reviewer for the comment. The observation that $Myc^{\Delta 2-540/\Delta 2-540}$ 3-D fetal cultures differ from the wildtype and yet retain comparable morphology is similar to our data from the adult mice where although the mutant intestine is morphologically indistinguishable from the wildtype during homeostasis, it is severely defective in regeneration and incapable of efficiently supporting tumor growth. **We have clarified this in the manuscript (Page9, lines 206-209).**

205-207: Clarify how the conclusions on proliferation, differentiation, and crypt-like structures were drawn from the data.

We thank the reviewer for the comment. Since the spheroids from $Myc^{\Delta 2-540/\Delta 2-540}$ can grow in 3D-cultures and we can identify several intestinal lineages from these cultures in scRNA-seq data (Supplementary figure S5), we concluded that $Myc^{\Delta 2-540/\Delta 2-540}$ fetal intestinal cells can proliferate and differentiate in 3D-cultures. However since we do not observe budding organoids with mature crypt-like structures in the first 10 days of culture we concluded that the mutant cells are not proficient in generating those. **We have modified our original statement in the text to**

“These results demonstrate that although fetal intestinal 3-D cultures can be propagated in almost complete absence of MYC, the adult repair and regenerative state is critically coupled to a MYC-dependent proliferation and mobilization of regenerative stem/progenitor cells.” **(Page11, lines 243-246).**

3G: the definition of specific clusters is unclear and the strategy for identification needs to be clarified. Particularly since the distinction of those clusters is not simply visible in the presented UMAPs.

We thank the reviewer for the comment. This was also pointed out by Ref#2. We have clarified the strategy for identifying the clusters shown in old Fig. 3G in **Figure legend for new Fig 5D and in Materials and Methods (Page20, lines 459-462)**.

Figure 4:

4D & Sup Fig. 5B: Include control images for all comparisons.

We thank the reviewer for the comment. **We have provided control images for Ki-67 and β -catenin staining in normal intestinal epithelium of $Apc^{Min/+}; Myc^{\Delta 2-540/\Delta 2-540}$ mice (revised Fig 6D)**. The β -catenin staining is nuclear in few cells at the base of the crypt while it is cytoplasmic in majority of the cells present within the crypts and villi. This is in agreement with published work in the field. **We have also revised old supplementary Fig 5B and provided an image of a wild-type mammary gland as control (revised supplementary Fig S6)**.

Leverage scRNA-seq data to support transcriptional changes in Fig. 4E and address previously discussed stem cell populations.

We thank the reviewer for the comment. Based on reviewer's suggestion, we have queried our scRNA-seq data to determine how the expression of *Lgr5*, *Myc*, *Ly6a* and *Notum* distributes over the regenerative stem cell subcluster (SSC_{reg}) during adult intestinal regeneration after irradiation and cancer. Although *Lgr5*, *Myc*, and *Ly6a* are expressed in SSC_{reg} cells from both irradiated WT intestine and $Apc^{Min/+}$ intestine, *Notum* expression specifically marks the SSC_{reg} stem cells from $Apc^{Min/+}$ mice, and is not expressed during regeneration. **We have now included this data in revised Fig 6. We have also clarified that the scRNA-seq data is from $Apc^{Min/+}$ and $Apc^{Min/+}; Myc^{\Delta 2-540/\Delta 2-540}$ mice at four months of age at which time $Apc^{Min/+}; Myc^{\Delta 2-540/\Delta 2-540}$ has almost no polyps/tumors. (Pages12, lines 262-264)**.

We also took the marker genes of the adult regenerative SSC cluster (SSC_{reg}) and checked their expression in stem cell subclusters from fetal 3-D organoid cultures. The adult regenerative set of genes were more expressed in the fetal $SSC2_F$, $SSC4_F$ and $SSC5_F$ subclusters which are enriched in $Myc^{\Delta 2-540/\Delta 2-540}$ cultures compared to the wild-type. This likely reflects the fetal reversion of adult stem cells during regeneration and cancer. However, unlike the fetal state that can support growth even with absent or

reduced *Myc* and *Ly6a* (Sca1) expression, the adult regenerative state appears to critically depend on *Myc* and *Ly6a* (Sca1) expressing cells. **We have discussed this in the manuscript (Page13-14, lines 305-315).**

4E: Question the choice of ISH over IHC; clarify the rationale behind focusing on specific changes, like Ly6a, which is critical yet not well addressed. Representative areas needs quantification, particularly since the bottom panel shows a rather cross sectioning area of the tumors.

We thank the reviewer for the comment. ISH was chosen because there are no good antibodies available for detecting *Lgr5* and MYC in mouse FFPE samples. We therefore used probes against *Myc* and *Lgr5* to determine the contribution of *Myc* and adult stem cells to tumors in *Apc^{Min/+}* and *Apc^{Min/+}; Myc^{Δ2-540/Δ2-540}* mice. Since *Ly6a* was altered in *Myc^{Δ2-540/Δ2-540}* intestine both during *in vivo* and *in vitro* regeneration, we also examined it in the tumor. *Notum* which provides a growth advantage to *Apc^{-/-}* stem cells over the neighboring wild-type counterparts, was selected to probe *Apc* mutant cells in and around the tumors. **We have now clarified this in the manuscript (Page12-13, lines 282-287).**

Regarding the representative areas we prefer to provide these images as qualitative images since the staining is very heterogeneous across the tumor. The tumor shown in the bottom panel of Fig 6E is rather large and we cannot provide better image for that without compromising resolution. However for the epithelium adjacent to the tumor **we have marked, in consultation with the pathologist, the structures where *Lgr5* staining can be observed in a pattern reminiscent of normal epithelium (New Fig 6E, red arrow heads marks one *Lgr5⁺* crypt across all panels of *Apc^{Min/+}; Myc^{Δ2-540/Δ2-540}* tumor). We have also provided images from an additional tumor boundary showing *Ly6a* (Sca1) and *Notum* expression where *Ly6a* (Sca1) is prominently expressed in the normal epithelium next to the tumor (new Supplementary Fig S7 with better oriented crypt-villi next to the tumor).**

4F & 4G: The purpose of the scRNA-seq controls is unclear; more relevant data could enhance understanding of the findings.

We thank the reviewer for the comment. Based on this and previous comment by the reviewer we have leveraged our scRNA-seq data for more relevant results and understanding of the findings. **We have provided a revised figure (new Fig 6) with additional results from a comparison of the stem cell subclusters from fetal organoid cultures and the regenerative stem cell subcluster of the adult (also see response to reviewers previous question on Fig 4E).**

Figure 5

Should be checked for typos and clarity of the findings needs to be enhance. It contains various aspects that have not been demonstrated in this paper like increase in mass and somatic stem cells. Again clarity with regards to the actual finding is difficult to define from the this figure.

We thank the reviewer for the comment. **We have corrected the typo and revised the figure (new Fig 7).** The figure now additionally shows the relationship between the cancer susceptibility region in humans and the mouse super-enhancer region deleted in our study. We have also clearly demarcated the different MYC requirement in adult vs fetal regeneration.

March 3, 2025

RE: Life Science Alliance Manuscript #LSA-2024-03090-TR

Prof. Jussi Taipale
Karolinska Institutet
Department of Medical Biochemistry and Biophysics
Biomedicum B9
Stockholm 171 77
Sweden

Dear Dr. Taipale,

Thank you for submitting your revised manuscript entitled "The 8q24 super-enhancer region is required for MYC-dependent mobilization of regenerative stem cells". We would be happy to publish your paper in Life Science Alliance pending final revisions necessary to meet our formatting guidelines.

- please be sure that the authorship listing and order is correct
- please add the X and Bluesky handles of your host institute/organization as well as your own or/and one of the authors in our system
- titles in the system and manuscript file must match
- please add callouts for Figure 5A; S1A-D; S2A-G and S6A-B to your main manuscript text
- the sequencing data should be made publicly accessible at this point

A. FINAL FILES:

B. MANUSCRIPT ORGANIZATION AND FORMATTING:

Sincerely,

March 14, 2025

RE: Life Science Alliance Manuscript #LSA-2024-03090-TRR

Prof. Jussi Taipale
Karolinska Institutet
Department of Medical Biochemistry and Biophysics
Biomedicum B9
Stockholm 171 77
Sweden

Dear Dr. Taipale,

Thank you for submitting your Research Article entitled "Shared requirement for MYC upstream super-enhancer region in tissue regeneration and cancer". It is a pleasure to let you know that your manuscript is now accepted for publication in Life Science Alliance. Congratulations on this interesting work.

DISTRIBUTION OF MATERIALS:

Again, congratulations on a very nice paper. I hope you found the review process to be constructive and are pleased with how the manuscript was handled editorially. We look forward to future exciting submissions from your lab.

Sincerely,
